# The Implicit Bias of Adam on Separable Data

**Chenyang Zhang**
Department of Statistics and Actuarial Science
School of Computing and Data Science
The University of Hong Kong
chyzhang@connect.hku.hk

**Difan Zou**
Department of Computer Science
School of Computing and Data Science
& Institute of Data Science
The University of Hong Kong
dzou@cs.hku.hk

**Yuan Cao**
Department of Statistics and Actuarial Science
School of Computing and Data Science
& Department of Mathematics
The University of Hong Kong
yuancao@hku.hk

## Abstract

Adam has become one of the most favored optimizers in deep learning problems. Despite its success in practice, numerous mysteries persist regarding its theoretical understanding. In this paper, we study the implicit bias of Adam in linear logistic regression. Specifically, we show that when the training data are linearly separable, the iterates of Adam converge towards a linear classifier that achieves the maximum $\ell_\infty$-margin in direction. Notably, for a general class of diminishing learning rates, this convergence occurs within polynomial time. Our result shed light on the difference between Adam and (stochastic) gradient descent from a theoretical perspective.

## 1 Introduction

Adam [25] is one of the most widely used optimization algorithms in deep learning. By entry-wisely adjusting the learning rate based on the magnitude of historical gradients, Adam has proven to be highly efficient in solving optimization tasks in machine learning. However, despite the remarkable empirical success of Adam, current theoretical understandings of Adam cannot fully explain its fundamental difference compared with other optimization algorithms.

It has been recently pointed out that the *implicit bias* [32, 40, 21] of an optimization algorithm is essential in understanding the performance of the algorithm in machine learning. In over-parameterized learning tasks where the training objective function may have infinitely many solutions, the implicit bias of an optimization algorithm characterizes how the algorithm prioritizes converging towards a specific optimum with particular structures and properties. Several recent works studied the implicit bias of Adam and other adaptive gradient methods. Specifically, [35] studied the implicit bias of AdaGrad, and showed that AdaGrad converges to a direction that can be characterized as the solution of a quadratic optimization problem related to the limit of preconditioners. However, their results cannot be extended to Adam. [45] showed that gradient descent with momentum (GDM) and its adaptive variants have the same implicit bias with gradient descent. This result is extended to the setting of training homogeneous models in [44]. However, the results in [45, 44] reply on a nonnegligible stability constant – when the gradient entries are minimized below the stability constant (which is by default $10^{-8}$ in Adam), adaptive gradient methods will essentially behave like gradient

38th Conference on Neural Information Processing Systems (NeurIPS 2024).

descent. Therefore, it remains an open question how Adam will behave under the more practical regime where stability constant is negligible.

We note that there exist several recent works studying variants of Adam without the stability constant. First of all, sign gradient descent, which can be regarded as Adam without momentum or stability constant, is usually considered a proxy of Adam in theoretical studies due to its ease of analysis. [14] studied the implicit bias of steepest descent with respect to different potentials and norms covering a variant of sign gradient descent, and demonstrated that sign gradient descent converges to the maximum $\ell_\infty$-margin solution. However, this result for sign gradient descent cannot cover Adam, as the momentum terms in the update of Adam are crucial. Besides, a more recent work [50] studied the implicit bias of AdamW without considering the stability constant. They showed that, if the iterates of AdamW converge, then the limiting point must be a KKT point of an optimization problem with $\ell_\infty$ constraints. However, the analysis of AdamW in [50] relies on a non-zero regularization parameter, and therefore cannot be extended to the study of Adam.

In this paper, we investigate the implicit bias of Adam. Specifically, let $\{(\mathbf{x}_i, y_i)\}_{i=1}^n \subset \mathbb{R}^d \times \{\pm 1\}$ be a training data set of a binary classification problem. We consider using Adam to train a linear model to minimize the empirical logistic loss (or exponential loss). Then our main results can be summarized as the following informal theorem:

**Theorem 1.1** (Simplified version of Theorem 4.5). Let $\{\eta_t\}_{t=0}^\infty$, $\{\mathbf{w}_t\}_{t=0}^\infty$ be the sequence of learning rates and iterates of Adam respectively. Suppose that the data set $\{(\mathbf{x}_i, y_i)\}_{i=1}^n$ is linearly separable, and that $\lim_{t\to} \eta_t = 0$, $\sum_{t=0}^\infty \eta_t = \infty$. Then under certain conditions, it holds that

$$\left| \min_{i \in [n]} \frac{\langle \mathbf{w}_t, y_i \cdot \mathbf{x}_i \rangle}{\|\mathbf{w}_t\|_\infty} - \gamma \right| \leq O\left( \frac{\sum_{\tau=0}^{t-1} \eta_\tau^{\frac{3}{2}} + \sum_{\tau=0}^{t-1} \eta_\tau e^{-\frac{\gamma}{4} \sum_{\tau'=0}^{\tau-1} \eta_{\tau'}}}{\sum_{\tau=0}^{t-1} \eta_\tau} \right), \quad \text{(for logistic loss)}$$

$$\left| \min_{i \in [n]} \frac{\langle \mathbf{w}_t, y_i \cdot \mathbf{x}_i \rangle}{\|\mathbf{w}_t\|_\infty} - \gamma \right| \leq O\left( \frac{\sum_{\tau=0}^{t-1} \eta_\tau^{\frac{3}{2}}}{\sum_{\tau=0}^{t-1} \eta_\tau} \right), \quad \text{(for exponential loss)}$$

where $\gamma := \max_{\|\mathbf{w}\|_\infty \leq 1} \min_{i \in [n]} \langle \mathbf{w}, y_i \cdot \mathbf{x}_i \rangle$ is the maximum $\ell_\infty$-margin on the training data set.

Theorem 1.1 shows that, for a general class of learning rate schedules, Adam will eventually achieve the maximum $\ell_\infty$-margin on the training data set.

- We demonstrate the implicit bias of Adam for solving linear logistic regression with linearly separable data. Specifically, we prove that Adam has an implicit bias towards a maximum $\ell_\infty$-margin solution when solving linear classification problems. Our result distinguishes Adam from (stochastic) gradient descent with/without momentum, whose implicit bias is towards the maximum $\ell_2$-margin solution.

- Our analysis of Adam covers a broad range of diminishing learning rate schedules. For $\eta_t = \Theta(t^{-a})$ with $a \in (0, 1]$, our result demonstrates the following convergence rate:

$$\left| \min_{i \in [n]} \frac{\langle \mathbf{w}_t, y_i \cdot \mathbf{x}_i \rangle}{\|\mathbf{w}_t\|_\infty} - \gamma \right| \leq \begin{cases} O\big(t^{-a/2}\big), & \text{if } a < 2/3; \\ O\big(t^{-1/3} \log t\big), & \text{if } a = 2/3; \\ O\big(t^{1-a}\big), & \text{if } 2/3 < a < 1; \\ O\big(1/\log t\big), & \text{if } a = 1. \end{cases}$$

Notably, when $a < 1$, the above rates indicate that the convergence towards the maximum $\ell_\infty$-margin occurs in polynomial time. This further differentiates Adam from (stochastic) gradient descent with/without momentum in terms of the convergence speed.

- Our result focuses on a particularly challenging setting where we ignore the "stability constant $\epsilon$" in the Adam algorithm. In practice, the stability constant is by default set as $\epsilon = 10^{-8}$, which is almost negligible throughout the optimization process. Therefore, by covering the setting without the stability constant, our theory matches the practical setting better. We demonstrate by simulation that our theory can also correctly characterize the implicit bias of Adam with the stability constant.

**Notation.** Given two sequences $\{x_n\}$ and $\{y_n\}$, we denote $x_n = O(y_n)$ if there exist some absolute constant $C_1 > 0$ and $N > 0$ such that $|x_n| \leq C_1 |y_n|$ for all $n \geq N$. Similarly, we denote $x_n = \Omega(y_n)$

if there exist $C_2 > 0$ and $N > 0$ such that $|x_n| \geq C_2|y_n|$ for all $n > N$. We say $x_n = \Theta(y_n)$ if $x_n = O(y_n)$ and $x_n = \Omega(y_n)$ both holds. We use $\widetilde{O}(\cdot), \widetilde{\Omega}(\cdot)$, and $\widetilde{\Theta}(\cdot)$ to hide logarithmic factors in these notations respectively. Moreover, we denote $x_n = \text{poly}(y_n)$ if $x_n = O(y_n^D)$ for some positive constant $D$, and $x_n = \text{polylog}(y_n)$ if $x_n = \text{poly}(\log(y_n))$. For two scalars $a$ and $b$, we denote $a \vee b = \max\{a, b\}$ and $a \wedge b = \min\{a, b\}$. For any $n \in \mathbb{N}_+$, we use $[n]$ to denote the set $\{1, 2, \cdots, n\}$. For any scalar $c \in \mathbb{R}$, $\lceil c \rceil$ denotes the smallest integer larger or equal to $c$ and $\lfloor c \rfloor$ denotes the largest integer smaller or equal to $c$. For a vector $\mathbf{a} \in \mathbb{R}^d$, $\mathbf{a}[k]$ denote its $k$-th entry. Finally, $\mathbf{e}_i \in \mathbb{R}^n$ denotes the $i$-th basis vector in $\mathbb{R}^n$.

## 2 Additional Related Work

**Theoretical analyses of Adam and its variants.** There has been a line of works studying the properties of Adam and its variants from different aspects. [37] pointed out that there exists simple convex objective functions which Adam may fail to minimize, and proposed a new variant of Adam, the AMSGrad algorithm, which enjoys convergence guarantees in convex optimization. [54, 10, 17, 33, 53, 18] established optimization guarantees of Adam and its variants in non-convex optimization. [29, 19] implemented variance reduction techniques in Adam and proposed new variants of Adam accordingly. [47, 52, 55, 56] studied the generalization performance of Adam and compared it with GD under different learning tasks. [27, 4, 6, 3, 5] tried to explain the performance of Adam by studying the connections between Adam and sign gradient descent. [51] explored the optimization trajectories of Adam from the $\ell_\infty$ geometry.

**Implicit bias.** Classic results [40, 21] demonstrated the iterates of GD will converge to the maximum $\ell_2$-margin solution in direction on linear logistic regression with linear separable datasets. [31] extended this result under stochastic settings. [14] explored the implicit bias of a general class of optimization methods, containing mirror descent and steepest descent. [23] proposed a primal-dual analysis and derived a faster convergence rate with a larger learning rate compared to [40, 21]. [48] explored the implicit bias of gradient descent at the 'edge of stability' regime, where the learning rate can be an arbitrarily large constant. [30, 22] showed that $q$-homogeneous neural network trained by GD will converge to a KKT point of maximum $\ell_2$-margin optimization problem. [8] established an implicit bias type result for the Lion [9] algorithm in its continuous-time form. There also exist numerous works studying the implicit bias for different problem setting, including matrix factorization models [16, 28, 2, 36], squared loss models [38, 1, 24], weight normalization and batch normalization [49, 7], deep linear neural networks [15, 20], two-layer neural networks [11, 34, 13, 42, 43, 26].

## 3 Problem Settings

We consider binary linear classification problems. Specifically, given $n$ training data points $\{(\mathbf{x}_i, y_i)\}_{i=1}^n$ where $\mathbf{x}_i \in \mathbb{R}^d$ and $y_i \in \{+1, -1\}$, we aim to find a coefficient vector $\mathbf{w}$ which minimizes the following empirical loss

$$\mathcal{R}(\mathbf{w}) = \frac{1}{n} \sum_{i=1}^n \ell(\langle \mathbf{w}, y_i \cdot \mathbf{x}_i \rangle), \tag{3.1}$$

where $\ell(\langle \mathbf{w}, y_i \cdot \mathbf{x}_i \rangle)$ is the loss function value on the data point $(\mathbf{x}_i, y_i)$. In this paper, we consider $\ell \in \{\ell_{\log}, \ell_{\exp}\}$, where $\ell_{\log}(z) = \log(1 + e^{-z})$ is the logistic loss function and $\ell_{\exp}(z) = e^{-z}$ is the exponential loss function. We consider using Adam to minimize (3.1). Denoting $\mathbf{m}_{-1} = \mathbf{v}_{-1} = \mathbf{0} \in \mathbb{R}^d$ and starting with initialization $\mathbf{w}_0$, Adam applies the following iterative formulas:

$$\mathbf{m}_t = \beta_1 \mathbf{m}_{t-1} + (1 - \beta_1) \cdot \nabla \mathcal{R}(\mathbf{w}_t), \tag{3.2}$$

$$\mathbf{v}_t = \beta_2 \mathbf{v}_{t-1} + (1 - \beta_2) \cdot \nabla \mathcal{R}(\mathbf{w}_t)^2, \tag{3.3}$$

$$\mathbf{w}_{t+1} = \mathbf{w}_t - \eta_t \frac{\mathbf{m}_t}{\sqrt{\mathbf{v}_t}}, \tag{3.4}$$

where $\beta_1, \beta_2 \in [0, 1)$ are the hyperparameters of Adam, and the square $(\cdot)^2$, square root $(\sqrt{\cdot})$ and division $(\div)$ above all denote entry-wise calculations.

Note that in practice, it is common to consider the variant $\mathbf{w}_{t+1} = \mathbf{w}_t - \eta_t \frac{\mathbf{m}_t}{\sqrt{\mathbf{v}_t}+\epsilon}$, where an additional term $\epsilon \approx 10^{-8}$ is added in (3.4) to improve stability. However, in our analysis, we do not consider such a term $\epsilon$. This is because in practice, one seldom run Adam until $\mathbf{v}_t$ is around the same level as $\epsilon$. However, by the nature of implicit bias, the result needs to cover infinitely many iterations, and the additional term $\epsilon$ will eventually significantly affect the result. In fact, a recent work [45] showed that when one considers such an additional $\epsilon$ term, Adam will be asymptotically equivalent to gradient descent. In comparison, in this paper, we will show that when ignoring $\epsilon$, Adam has a unique implicit bias that is different from gradient descent.

## 4 Main Results

In this section, we present our main result on the implicit bias of Adam in linear classification problems. We first introduce several assumptions.

**Assumption 4.1.** There exists $\mathbf{w} \in \mathbb{R}^d$ such that $\langle \mathbf{w}, y_i \cdot \mathbf{x}_i \rangle > 0$ for all $i \in [n]$.

Assumption 4.1 is a standard assumption in the study of implicit bias of linear models [40, 14, 31, 21, 23, 45, 48]. It can be easily satisfied in the over-parameterized setting where $d \geq n$. With the linear separability assumption, we can further define the maximum $\ell_\infty$-margin:

$$\gamma = \max_{\|\mathbf{w}\|_\infty \leq 1} \min_{i \in [n]} \langle \mathbf{w}, y_i \cdot \mathbf{x}_i \rangle. \tag{4.1}$$

We also make the following assumption on the initialization $\mathbf{w}_0$.

**Assumption 4.2.** The initialization $\mathbf{w}_0$ of Adam satisfies that for all $k \in [d]$, $\nabla \mathcal{R}(\mathbf{w}_0)[k]^2 \geq \rho$.

Assumption 4.2 ensures that at every finite iteration, the entries of $\mathbf{v}_t$ are strictly positive. We remark that this is a mild assumption: if $\mathbf{x}_i$, $i \in [n]$ are generated from a continuous, non-degenerate distribution, then regardless of the choice of $\mathbf{w}_0$, $\nabla \mathcal{R}(\mathbf{w}_0)[k] \neq 0$ with probability 1. Moreover, $\rho$ will only appear in our results in the form of $\log(1/\rho)$, and therefore, even if $\rho$ is small, it will not significantly hurt the convergence rates. A similar assumption has also been considered in [50].

**Assumption 4.3.** $\{\eta_t\}_{t=1}^\infty$ are decreasing in $t$, and satisfy $\sum_{t=0}^\infty \eta_t = \infty$, $\lim_{t \to \infty} \eta_t = 0$.

Assumption 4.3 is a mild and standard assumption of the learning rates $\{\eta_t\}_{t=0}^\infty$ that is commonly considered in the general optimization literature. It has also been considered in recent studies of Adam and its variants [12, 19, 50].

**Assumption 4.4.** For all $\beta \in (0,1)$ and $c_1 > 0$, there exist $t_1 \in \mathbb{N}_+$ and $c_2 > 0$ that only depend on $\beta, c_1$, such that $\sum_{\tau=0}^t \beta^\tau \left( e^{c_1 \sum_{\tau'=1}^\tau \eta_{t-\tau'}} - 1 \right) \leq c_2 \eta_t$ for all $t \geq t_1$.

Although Assumption 4.4 seems non-trivial, we claim it is a fairly mild assumption. In fact, for both small fixed learning rate $\eta_t = \eta$, and decay learning rate $\eta_t = (t+2)^{-a}$ with $a \in (0,1]$, Assumption 4.4 always hold. We formally prove this result in Lemma C.1 in the appendix.

Now, we state our main theorem about the implicit bias about Adam as follows.

**Theorem 4.5.** Let $\{\mathbf{w}_t\}_{t=0}^\infty$ be the iterates of Adam in (3.2)-(3.4) with $\beta_1 \leq \beta_2$. In addition, let $\gamma$ be defined in (4.1) and $B := \max_{i \in [n]} \|\mathbf{x}_i\|_1$. Then under Assumptions 4.1, 4.2, 4.3 and 4.4, there exists $t_0 = t_0(n, d, \beta_1, \beta_2, \gamma, B, \rho, \mathbf{w}_0)$ such that

- If $\ell = \ell_{\exp}$, then for all $t \geq t_0$,

$$\mathcal{R}(\mathbf{w}_t) \leq \frac{\log 2}{n} \cdot e^{-\frac{\gamma}{2} \sum_{\tau=t_0}^{t-1} \eta_\tau}, \text{ and } \left| \min_{i \in [n]} \frac{\langle \mathbf{w}_t, y_i \cdot \mathbf{x}_i \rangle}{\|\mathbf{w}_t\|_\infty} - \gamma \right| \leq O\left( \frac{\sum_{\tau=0}^{t_0-1} \eta_\tau + d \sum_{\tau=t_0}^{t-1} \eta_\tau^{\frac{3}{2}}}{\sum_{\tau=0}^{t-1} \eta_\tau} \right).$$

- If $\ell = \ell_{\log}$, then for all $t \geq t_0$,

$$\mathcal{R}(\mathbf{w}_t) \leq \frac{\log 2}{n} \cdot e^{-\frac{\gamma}{4} \sum_{\tau=t_0}^{t-1} \eta_\tau},$$

and

$$\left| \min_{i \in [n]} \frac{\langle \mathbf{w}_t, y_i \cdot \mathbf{x}_i \rangle}{\|\mathbf{w}_t\|_\infty} - \gamma \right| \leq O\left( \frac{\sum_{\tau=0}^{t_0-1} \eta_\tau + d \sum_{\tau=t_0}^{t-1} \eta_\tau^{\frac{3}{2}} + \sum_{\tau=t_0}^{t-1} \eta_\tau e^{-\frac{\gamma}{4} \sum_{\tau'=t_0}^{\tau-1} \eta_{\tau'}}}{\sum_{\tau=0}^{t-1} \eta_\tau} \right),$$

where we use $O(\cdot)$ to omit factors that only depend on $\beta_1, \beta_2, \gamma, B$.

Theorem 4.5 implies that Adam can minimize the loss function to zero, and that the normalized $\ell_\infty$-margin achieved by Adam will eventually converge to the maximum $\ell_\infty$-margin of the training data set. To address general learning rate schedules, we do not specify a particular convergence rate for either the loss or the margin, nor do we provide an exact formula for $t_0$. However, it can be easily verified that $\mathcal{R}(\mathbf{w}_t) \leq O\big(e^{-\gamma t^{1-a}/4(1-a)}\big)$ when $\eta_t = (t+2)^{-a}$ with $a < 1$. This loss convergence rate of Adam is much faster than that of (stochastic) gradient descent (with momentum) given a fixed small learning rate, which is of order $O(1/t)$ [40, 31, 45]. In addition, we have $t_0 = \mathrm{poly}[n, d, (1-\beta_1)^{-1}, (1-\beta_2)^{-1}, \gamma^{-1}, B, \log(1/\rho), \mathcal{R}(\mathbf{w}_0)]$ when $\eta_t = (t+2)^{-a}$ with $a < 1$, and we defer the derivation details to Appendix B.2. Regarding margin convergence, we will give a set of detailed convergence rate results for different learning rate schedules in Corollary 4.7.

According to Theorem 4.5, the nature of Adam is vastly different from (stochastic) gradient descent from the perspective of implicit bias: Adam maximizes the $\ell_\infty$-margin, while existing works have demonstrated that (stochastic) gradient descent maximizes the $\ell_2$-margin [40, 31, 21]. Compared with existing works on the implicit bias of adaptive gradient methods [35, 45, 50], our result demonstrates a novel type of implicit bias with accurate convergence rates, which can not been covered in the previous results. Notably, [45] showed that, if a stability constant $\epsilon$ is added, i.e., (3.4) is replaced by $\mathbf{w}_{t+1} = \mathbf{w}_t - \eta_t \frac{\mathbf{m}_t}{\sqrt{\mathbf{v}_t}+\epsilon}$, then Adam will eventually be equivalent to gradient descent and will converge to the maximum $\ell_2$-margin solution. However, the analysis in [45] relies on a positive $\epsilon$: their proof is based the fact that after a large number of iterations, the entries of $\mathbf{v}_t$ will eventually be much smaller than $\epsilon$, and the update of Adam will be similar to gradient descent with momentum. In our analysis, we are able to cover the setting where $\epsilon = 0$, and our result demonstrates that studying the setting without $\epsilon$ is essential, as the implicit bias is completely different. In Section 5, we will demonstrate by experiments that our setting matches the practical observations better.

As we have discussed, Theorem 4.5 implies the convergence of the normalized $\ell_\infty$-margin of Adam iterates towards the maximum $\ell_\infty$-margin. Since the results cover very general learning rates, the convergence rates are presented in rather complicated formats. However, based on the assumption that $\sum_{t=0}^\infty \eta_t = \infty$, $\lim_{t\to\infty} \eta_t = 0$, we can immediately conclude the following simplified result by the Stolz–Cesàro theorem (see Theorem C.8 in the appendix).

**Corollary 4.6.** Under the same conditions in Theorem 4.5, it holds that

$$\lim_{t\to\infty} \min_{i\in[n]} \frac{\langle \mathbf{w}_t, y_i \cdot \mathbf{x}_i \rangle}{\|\mathbf{w}_t\|_\infty} = \max_{\|\mathbf{w}\|_\infty \leq 1} \min_{i\in[n]} \langle \mathbf{w}, y_i \cdot \mathbf{x}_i \rangle.$$

If there exists a unique maximum $\ell_\infty$-margin solution $\mathbf{w}^* = \mathrm{argmax}_{\|\mathbf{w}\|_\infty \leq 1} \min_{i\in[n]} \langle \mathbf{w}, \mathbf{x}_i \rangle$, then we have $\lim_{t\to\infty} \frac{\mathbf{w}_t}{\|\mathbf{w}_t\|_\infty} = \mathbf{w}^*$.

We can also investigate the convergence rates of the $\ell_\infty$-margin with specific learning rates. The results are summarized in the following Corollary.

**Corollary 4.7.** Consider $\eta_t = (t+2)^{-a}$ with $a \in (0,1]$. Denote by $\mathbf{w}_t^{\exp}$ and $\mathbf{w}_t^{\log}$ the iterates of Adam for $\ell = \ell_{\exp}$ and $\ell = \ell_{\log}$ respectively. Suppose that $\beta_1 \leq \beta_2$ and Adam starts with initialization $\mathbf{w}_0$. Let $B := \max_{i\in[n]} \|\mathbf{x}_i\|_1$. Then under Assumptions 4.1 and 4.2, there exists $t_0 = t_0(n, d, \beta_1, \beta_2, \gamma, B, \rho, \mathbf{w}_0)$ such that for all $t \geq t_0$, the following results hold:

• If $a < \frac{2}{3}$,

$$\left| \min_{i\in[n]} \frac{\langle \mathbf{w}_t^{\exp}, y_i \cdot \mathbf{x}_i \rangle}{\|\mathbf{w}_t^{\exp}\|_\infty} - \gamma \right|, \left| \min_{i\in[n]} \frac{\langle \mathbf{w}_t^{\log}, y_i \cdot \mathbf{x}_i \rangle}{\|\mathbf{w}_t^{\log}\|_\infty} - \gamma \right| \leq O\left( \frac{d}{t^{a/2}} \right).$$

• If $a = \frac{2}{3}$,

$$\left| \min_{i\in[n]} \frac{\langle \mathbf{w}_t^{\exp}, y_i \cdot \mathbf{x}_i \rangle}{\|\mathbf{w}_t^{\exp}\|_\infty} - \gamma \right| \leq O\left( \frac{d \cdot \log t + \log n + \log \mathcal{R}(\mathbf{w}_0) + [\log(1/\rho)]^{1/3}}{t^{1/3}} \right),$$

$$\left| \min_{i\in[n]} \frac{\langle \mathbf{w}_t^{\log}, y_i \cdot \mathbf{x}_i \rangle}{\|\mathbf{w}_t^{\log}\|_\infty} - \gamma \right| \leq O\left( \frac{d \cdot \log t + nd + n\mathcal{R}(\mathbf{w}_0) + [\log(1/\rho)]^{1/3}}{t^{1/3}} \right).$$

- If $\frac{2}{3} < a < 1$,

$$\left| \min_{i \in [n]} \frac{\langle \mathbf{w}_t^{\exp}, y_i \cdot \mathbf{x}_i \rangle}{\|\mathbf{w}_t^{\exp}\|_\infty} - \gamma \right| \leq O\left( \frac{d + \log n + \log \mathcal{R}(\mathbf{w}_0) + [\log(1/\rho)]^{1-a}}{t^{1-a}} \right),$$

$$\left| \min_{i \in [n]} \frac{\langle \mathbf{w}_t^{\log}, y_i \cdot \mathbf{x}_i \rangle}{\|\mathbf{w}_t^{\log}\|_\infty} - \gamma \right| \leq O\left( \frac{d + nd^{\frac{2(1-a)}{a}} + n\mathcal{R}(\mathbf{w}_0) + [\log(1/\rho)]^{1-a}}{t^{1-a}} \right).$$

- If $a = 1$,

$$\left| \min_{i \in [n]} \frac{\langle \mathbf{w}_t^{\exp}, y_i \cdot \mathbf{x}_i \rangle}{\|\mathbf{w}_t^{\exp}\|_\infty} - \gamma \right| \leq O\left( \frac{d + \log n + \log \mathcal{R}(\mathbf{w}_0) + \log\log(1/\rho)}{\log t} \right),$$

$$\left| \min_{i \in [n]} \frac{\langle \mathbf{w}_t^{\log}, y_i \cdot \mathbf{x}_i \rangle}{\|\mathbf{w}_t^{\log}\|_\infty} - \gamma \right| \leq O\left( \frac{d + n\log d + n\mathcal{R}(\mathbf{w}_0) + \log\log(1/\rho)}{\log t} \right).$$

Corollary 4.7 comprehensively presents the convergence rate of the $\ell_\infty$-margin for different learning rates. It also indicates that the margin convergence rates for $\ell_{\exp}$ and $\ell_{\log}$ are of the same order of $t$. Notably, for $a < 1$, the normalized $\ell_\infty$-margin converges in polynomial time. This clearly distinguishes Adam from (stochastic) gradient descent with/without momentum, for which the normalized $\ell_2$-margin converges at a speed $O(1/\log t)$ [40, 20, 45]. We note that a recent work [46] proposed a novel algorithm named progressive rescaling gradient descent that can maximize the margin at an exponential rate. Here our focus is different from [46]: our purpose is not to propose new algorithms to achieve better convergence rates, but is to theoretically study the properties of the classic Adam algorithm. We would also like to remark that, although Corollary 4.7 seemingly indicates that $\eta_t = (t + 2)^{-2/3}$ is the learning rate schedule with the fastest convergence rate, it does not mean that $\eta_t = (t + 2)^{-2/3}$ always converge faster than the other learning rate schedules in all learning tasks. The bounds in Corollary 4.7 are derived under the worst cases, and in practice, we can frequently observe that the margins all converge faster than the bounds in the corollary.

## 5 Experiments

In this section, we conduct numerical experiments to verify our theoretical conclusions. We set the sample size $n = 50$, and dimension $d = 50$. Then the data set $\{(\mathbf{x}_i, y_i)\}$ are generated as follows:

1. $\mathbf{x}_i$, $i \in [n]$ are independently generated from $N(\mathbf{0}, \mathbf{I})$.

2. $y_i$, $i \in [n]$ are independently generated from as $+1$ or $-1$ with equal probability.

Note that for data sets generated following the procedure above, Assumption 4.1 almost surely holds. We can also apply standard convex optimization to calculate the maximum $\ell_\infty$-margin $\gamma$. In order to make a clearer comparison between Adam and GD, we generate 10 independent sets of data, and we select the dataset with the most significant difference in the directions of the maximum $\ell_2$-margin solution and maximum $\ell_\infty$-margin solution. We then run the experiments on this selected data set. Throughout our experiments, for gradient descent with momentum, we set the momentum parameter as $\beta_1 = 0.9$, and for Adam, we set $\beta_1 = 0.9$, $\beta_2 = 0.99$. All these hyper-parameter setups are common in practice. All optimization algorithms are initialized with standard Gaussian distribution, and are run for $10^6$ iterations.

We first run GD, GDM, Adam without the stability constant, and Adam with stability constant $\epsilon = 10^{-8}$ to train a linear model minimizing the logistic loss, and compare their normalized $\ell_\infty$-margin and normalized $\ell_2$-margin. The results are given in Figure 1. We can see that the normalized $\ell_\infty$-margins of Adam, both with and without $\epsilon$, converge to the maximum $\ell_\infty$-margin, whereas the normalized $\ell_\infty$-margins of GD and GDM do not. In contrast, the normalized $\ell_2$-margins of GD and GDM converge to the maximum $\ell_2$-margin, while the $\ell_2$-margins of Adam, both with and without $\epsilon$, do not. By comparing the curves of Adam with and without $\epsilon$, we find that they behave similarly and their convergence remains highly stable. This justifies our theoretical setting where we ignore the stability constant in Adam, and demonstrate that our maximum $\ell_\infty$-margin implicit bias result

derived without $\epsilon$ characterizes the practical behaviour of Adam more accurately compared with the maximum $\ell_2$-margin result for Adam with $\epsilon$ in [45].

We also run a set of experiments to demonstrate the polynomial time convergence rate of the $\ell_\infty$-margin. We run experiments on Adam with learning rates $\eta_t = \Theta(t^{-a})$ for $a \in \{0.3, 0.5, 0.7, 1\}$, and report the log-log plots in Figure 2, where we perform the experiments for Adam with/without the stability constant separately. In the log-log plot, we observe that after a certain number of iterations, curves for $a < 1$ almost appear as straight lines, suggesting that the normalized $\ell_\infty$-margin converges in polynomial time for $a < 1$, while the curve for $a = 1$ exhibits logarithmic behavior, indicating the normalized $\ell_\infty$-margin converges logarithmically in $t$ for $a = 1$. Similarly to the previous observations, there is still no significant distinction between Adam with and without $\epsilon$, further demonstrating that our theoretical setting, which disregards $\epsilon$, is reasonable. We also note that in Figure 2, the margin achieved by Adam with $\eta_t = \Theta(t^{-0.3})$ converges the fastest. However, as we have commented in Section 4, different learning rate schedules may perform differently on different data sets, and it is not necessarily true that $\eta_t = \Theta(t^{-0.3})$ is always the best learning rate schedule.

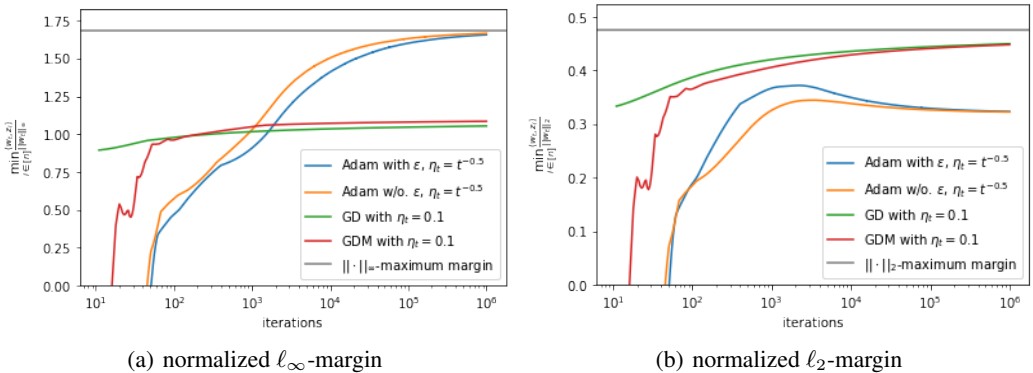

(a) normalized $\ell_\infty$-margin

(b) normalized $\ell_2$-margin

Figure 1: Normalized $\ell_\infty$-margins and $\ell_2$-margins achieved by GD, GDM, and Adam with/without the stability constant $\epsilon$ during training. (a) gives the results of normalized $\ell_\infty$-margins, while (b) shows the results of normalized $\ell_2$-margins.

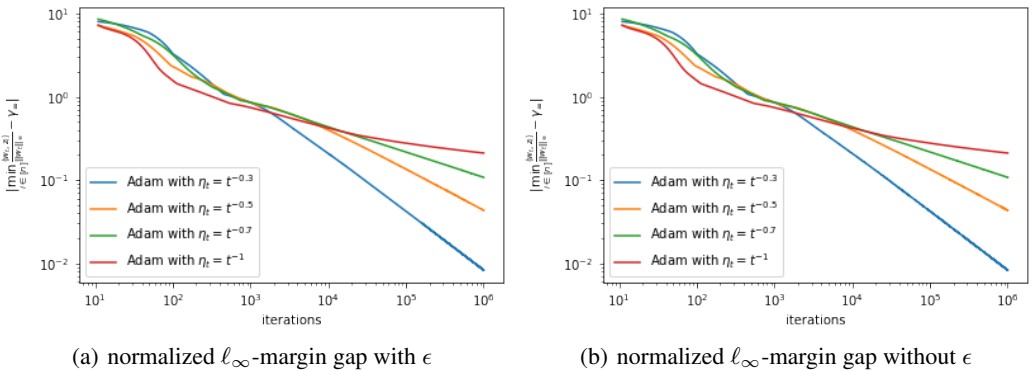

(a) normalized $\ell_\infty$-margin gap with $\epsilon$

(b) normalized $\ell_\infty$-margin gap without $\epsilon$

Figure 2: Log-log plots of the normalized $\ell_\infty$-margin gaps $|\min_{i \in [n]} \langle \mathbf{w}_t, y_i \cdot \mathbf{x}_i \rangle / \|\mathbf{w}_t\|_\infty - \gamma|$ versus training iterations. (a) presents the results for Adam with the stability constant $\epsilon$, and (b) presents the results for Adam without the stability constant $\epsilon$.

## 6 Proof Sketch for Theorem 4.5

In this section, we explain how we establish the convergence of the $\ell_\infty$-margin of linear models trained by Adam, and provide the sketch proof of Theorem 4.5. For simplicity, here we focus on the case $\ell = \ell_{\exp}$. The proof for $\ell = \ell_{\log}$ is almost the same.

We first introduce several notations. Define

$$\mathcal{G}(\mathbf{w}) = -\frac{1}{n}\sum_{i=1}^{n} \ell'(\langle \mathbf{w}, y_i \cdot \mathbf{x}_i \rangle).$$

Then for $\ell \in \{\ell_{\exp}, \ell_{\log}\}$, it is clear that $\mathcal{G}(\mathbf{w}) > 0$ for all $\mathbf{w} \in \mathbb{R}^d$. In the following, we will show that $\mathcal{G}(\mathbf{w})$ plays a key role in the convergence and implicit bias analysis.

**Step 1. Accurate characterizations of the first and second moments.** Adam algorithm is defined based on the first and second moments $\mathbf{m}_t$ and $\mathbf{v}_t$, which are calculated as exponential moving averages of the historical gradients and squared gradients respectively. A key challenge in studying Adam is to accurately characterize each entry of $\mathbf{m}_t$ and $\mathbf{v}_t$ throughout training. We present the following lemma.

**Lemma 6.1.** Under the same condition in Theorem 4.5, there exists $t_1 = t_1(\beta_1, \beta_2, B)$ such that

$$\left| \mathbf{m}_t[k] - (1 - \beta_1^{t+1}) \cdot \nabla \mathcal{R}(\mathbf{w}_t)[k] \right| \leq c_m \eta_t \mathcal{G}(\mathbf{w}_t),$$

$$\left| \sqrt{\mathbf{v}_t[k]} - \sqrt{1 - \beta_2^{t+1}} \cdot \left| \nabla \mathcal{R}(\mathbf{w}_t)[k] \right| \right| \leq c_v \sqrt{\eta_t} \mathcal{G}(\mathbf{w}_t)$$

for all $t > t_1$ and $k \in [d]$, where $c_m$ and $c_v$ are constants that only depend on $\beta_1$, $\beta_2$ and $B$.

Since $\eta_t$, $\beta_1^{t+1}$ and $\beta_2^{t+1}$ all decrease to zero as $t$ increases, Lemma 6.1 implies that after a sufficient number of iterations, the entries of $\mathbf{m}_t$ and $\mathbf{v}_t$ will be close to the corresponding entries of $\nabla \mathcal{R}(\mathbf{w}_t)$ and $|\nabla \mathcal{R}(\mathbf{w}_t)|$ respectively. Notably, the term $\mathcal{G}(\mathbf{w}_t)$ also appears in the bounds. In fact, deriving such bounds with the factor $\mathcal{G}(\mathbf{w}_t)$ is essential to enable our implicit bias analysis: when the algorithm converges, by definition, $\mathcal{G}(\mathbf{w}_t)$ will also decrease to zero, which implies that the bounds with the factor $\mathcal{G}(\mathbf{w}_t)$ are strictly tighter than the bounds without $\mathcal{G}(\mathbf{w}_t)$. Lemma 6.1 is one of our key technical contributions.

**Step 2. $\mathcal{R}(\mathbf{w}_t)$ starts to decrease after a fixed number of iterations.** Based on Lemma 6.1, we can analyze the convergence of $\mathcal{R}(\mathbf{w}_t)$. Specifically, we can show that, after a fixed number of iterations, the training loss function will start to decrease. This result is summarized in the following lemma.

**Lemma 6.2.** Under the same condition in Theorem 4.5, there exist $t_1 = t_1(\beta_1, \beta_2, B)$ such that for all $t > t_1$, it holds that

$$\mathcal{R}(\mathbf{w}_{t+1}) \leq \mathcal{R}(\mathbf{w}_t) - \eta_t \gamma \cdot \left( 1 - C_1 \beta_1^{t/2} - C_2 d \cdot \left( \eta_t^{\frac{1}{2}} + \eta_t \right) \right) \cdot \mathcal{G}(\mathbf{w}_t),$$

where $C_1, C_2$ only depend on $\beta_1, \beta_2, B$.

Note that by definition, $\mathcal{G}(\mathbf{w}) > 0$ for all $\mathbf{w} \in \mathbb{R}^d$. Therefore, Lemma 6.2 implies that $\mathcal{R}(\mathbf{w}_t)$ starts to decrease after a fixed number of iterations, and gives a bound on the decreasing speed. We remark that the proof of Lemma 6.2 is highly non-trivial. Although we have related $\mathbf{m}_t$ and $\mathbf{v}_t$ to the loss gradient $\nabla \mathcal{R}(\mathbf{w}_t)$ in Lemma 6.1, the fact that $\mathbf{w}_{t+1}$ is updated according to the entry-wise ratio $\mathbf{m}_t/\sqrt{\mathbf{v}_t}$ still introduces challenges: under our problem setting, it is entirely possible that at a certain iteration, a certain entry of $\nabla \mathcal{R}(\mathbf{w}_t)$ will exactly equal zero. In this case, the results in Lemma 6.1 can not directly lead to any conclusions about the ratio $\mathbf{m}_t/\sqrt{\mathbf{v}_t}$. In our proof, we implement a careful inequality that also takes the historical values of $\nabla \mathcal{R}(\mathbf{w}_t)$ into consideration.

**Step 3. Lower bound for un-normalized margin.** The proof of the implicit bias towards maximum $\ell_\infty$-margin also relies on a tight analysis on the un-normalized margin $\min_{i \in [n]} \langle \mathbf{w}_t, y_i \cdot \mathbf{x}_i \rangle$ during training. We have the following lemma providing a lower bound on the un-normalized margin.

**Lemma 6.3.** Under the same condition in Theorem 4.5, if there exists $t_0$ such that $\mathcal{R}(w_t) \leq \frac{\log 2}{n}$ for all $t \geq t_0$, then it holds that

$$\min_{i \in [n]} \langle \mathbf{w}_t, y_i \cdot \mathbf{x}_i \rangle \geq \gamma \sum_{\tau=t_0}^{t-1} \eta_\tau \cdot \frac{\mathcal{G}(\mathbf{w}_\tau)}{\mathcal{R}(\mathbf{w}_\tau)} - C_3 d \left( \sum_{\tau=t_0}^{t-1} \eta_\tau^{\frac{3}{2}} + \sum_{\tau=t_0}^{t-1} \eta_\tau^2 \right) - C_4$$

for all $t \geq t_0$, where $C_3, C_4$ only depend on $\beta_1, \beta_2, B$.

Note that this lower bound contains a negative term $-C_3 d \left( \sum_{\tau=t_0}^{t-1} \eta_\tau^{3/2} + \sum_{\tau=t_0}^{t-1} \eta_\tau^2 \right)$. Under our (mild) assumptions on the learning rates, it is entirely possible that $\sum_{\tau=t_0}^{\infty} \eta_\tau^{3/2}, \sum_{\tau=t_0}^{\infty} \eta_\tau^2 =$

$+\infty$ and thus the negative term in the lower bound may go to $-\infty$. However, we can show that $\lim_{t\to\infty} \mathcal{G}(\mathbf{w}_t)/\mathcal{R}(\mathbf{w}_t) = 1$ (in fact, for exponential loss, it is obvious that $\mathcal{G}(\mathbf{w}_t)/\mathcal{R}(\mathbf{w}_t) = 1$). Therefore, after a fixed number of iterations, the positive term in the lower bound will dominate, and Lemma 6.3 gives a non-trivial bound. The strength of this lemma lies in its applicability to very general learning rates $\{\eta_t\}_{t=1}^\infty$.

**Step 4. An upper bound of $\|\mathbf{w}_t\|_\infty$.**

In Lemma 6.3, we have obtained a lower bound of the un-normalized margin. However, to show the convergence of the $\ell_\infty$-normalized margin, we also need to establish a tight upper bound of $\|\mathbf{w}_t\|_\infty$. We present this result in the following lemma, which is inspired by Lemma 4.2 in [50].

**Lemma 6.4.** Suppose that the same conditions in Theorem 4.5 hold. There exist $C_5, C_6$ that only depend on $\beta_1, \beta_2, B$, such that the following result hold: if there exists $t_0 > \log(1/\rho)$ such that $\mathcal{R}(w_t) \leq \frac{1}{\sqrt{B^2 + C_5 \eta_0}}$ for all $t \geq t_0$, then $\|\mathbf{w}_t\|_\infty \leq \sum_{\tau=t_0}^{t-1} \eta_\tau + C_6 \sum_{\tau=0}^{t_0-1} \eta_\tau$ for all $t > t_0$.

Lemma 6.4 gives an upper bound of $\|\mathbf{w}_t\|_\infty$ which mainly depends on $\sum_{\tau=t_0}^{t-1} \eta_\tau$. Note that Lemma 6.3 also gives a lower bound of the un-normalized margin which mainly depends on $\sum_{\tau=t_0}^{t-1} \eta_\tau \mathcal{G}(\mathbf{w}_\tau)/\mathcal{R}(\mathbf{w}_\tau)$. These two lemmas will be combined to derive the convergence of the normalized margin.

**Step 5. Finalizing the proof.** Finally, based on the lemmas established in the previous steps, we can prove Theorem 4.5. We also need the following utility lemma provided by [56].

**Lemma 6.5** (Lemma A.2 in [56]). For Adam iterations defined in (3.2)-(3.4) with $\beta_1 \leq \beta_2$ and let $\alpha = \sqrt{\frac{\beta_2(1-\beta_1)^2}{(1-\beta_2)(\beta_2-\beta_1^2)}}$, then $\mathbf{m}_t[k] \leq \alpha \cdot \sqrt{\mathbf{v}_t[k]}$ for all $k \in [d]$.

We are now ready to prove Theorem 4.5 for the case $\ell = \ell_{\exp}$.

*Proof of Theorem 4.5.* By Lemma 6.2, there exists $t_2 = t_2(d, \beta_1, \beta_2, \gamma, B)$ such that

$$\mathcal{R}(\mathbf{w}_{t+1}) \leq \mathcal{R}(\mathbf{w}_t) - \frac{\gamma \eta_t}{2} \mathcal{G}(\mathbf{w}_t) \tag{6.1}$$

for all $t \geq t_2$. Note that for $\ell = \ell_{\exp}$, by definition we have $\mathcal{G}(\mathbf{w}_t) = \frac{1}{n} \sum_{i=1}^n \exp(-\langle \mathbf{w}_t, y_i \cdot \mathbf{x}_i \rangle) = \mathcal{R}(\mathbf{w}_t)$. Therefore, for all $t > t_2$, (6.1) can be re-written as

$$\mathcal{R}(\mathbf{w}_{t+1}) \leq \left(1 - \frac{\gamma \eta_t}{2}\right) \cdot \mathcal{R}(\mathbf{w}_t) \leq \mathcal{R}(\mathbf{w}_t) \cdot e^{-\frac{\gamma \eta_t}{2}} \leq \mathcal{R}(\mathbf{w}_{t_2}) \cdot e^{-\frac{\gamma \sum_{\tau=t_2}^t \eta_\tau}{2}}.$$

Although $\ell_{\exp}$ is not Lipschitz continuous over $\mathbb{R}$, we have $\mathcal{R}(\mathbf{w}_{t_2}) \leq \mathcal{R}(\mathbf{w}_0) \cdot e^{\alpha B \sum_{\tau=0}^{t_2-1} \eta_\tau}$ by Lemma 6.5 and triangle inequality. Letting $\mathcal{R}_0 = \min\{\frac{\log 2}{n}, \frac{1}{\sqrt{B^2 + C_5 \eta_0}}\}$ and $t_0 = t_0(n, d, \beta_1, \beta_2, \gamma, B, \rho, \mathbf{w}_0)$ be the first time such that $\sum_{\tau=t_2}^{t_0-1} \eta_\tau \geq \frac{2\alpha B}{\gamma} \sum_{\tau=0}^{t_2-1} \eta_\tau + \frac{2 \log \mathcal{R}(\mathbf{w}_0) - 2 \log \mathcal{R}_0}{\gamma}$ and $t_0 \geq -\log \rho$. By such definition of $t_0$, we can derive that for all $t \geq t_0$,

$$\mathcal{R}(\mathbf{w}_t) \leq \mathcal{R}(\mathbf{w}_{t_2}) \cdot e^{-\frac{\gamma \sum_{\tau=t_0}^t \eta_\tau}{2}} \cdot e^{-\frac{\gamma \sum_{\tau=t_2}^{t_0-1} \eta_\tau}{2}} \leq \mathcal{R}_0 \cdot e^{-\frac{\gamma \sum_{\tau=t_0}^t \eta_\tau}{2}},$$

which proves the bound on $\mathcal{R}(\mathbf{w}_t)$. Since $t_0$ satisfies all the conditions in Lemmas 6.3 and 6.4, by Lemmas 6.3, 6.4 and the fact that $\mathcal{G}(\mathbf{w}_\tau) = \mathcal{R}(\mathbf{w}_\tau)$ for exponential loss, we have

$$\frac{\langle \mathbf{w}_t, y_i \cdot \mathbf{x}_i \rangle}{\|\mathbf{w}_t\|_\infty} \geq \frac{\gamma \sum_{\tau=t_0}^{t-1} \eta_\tau - C_3 d\left(\sum_{\tau=t_0}^{t-1} \eta_\tau^{\frac{3}{2}} + \sum_{\tau=t_0}^{t-1} \eta_\tau^2\right) - C_4}{\sum_{\tau=t_0}^{t-1} \eta_\tau + C_6 \sum_{\tau=0}^{t_0-1} \eta_\tau}$$

for all $i \in [n]$, where $C_3, C_4$ and $C_6$ are constants solely depending on $\beta_1, \beta_2$ and $B$. Now by definition, we have $\gamma \geq \min_{i \in [n]} \langle \mathbf{w}_t, y_i \cdot \mathbf{x}_i \rangle / \|\mathbf{w}_t\|_\infty$. Therefore, we have

$$\left| \min_{i \in [n]} \frac{\langle \mathbf{w}_t, y_i \cdot \mathbf{x}_i \rangle}{\|\mathbf{w}_t\|_\infty} - \gamma \right| \leq \frac{\gamma C_6 \sum_{\tau=0}^{t_0-1} \eta_\tau + C_3 d\left(\sum_{\tau=t_0}^{t-1} \eta_\tau^{\frac{3}{2}} + \sum_{\tau=t_0}^{t-1} \eta_\tau^2\right) + C_4}{\sum_{\tau=t_0}^{t-1} \eta_\tau + C_6 \sum_{\tau=0}^{t_0-1} \eta_\tau}$$

$$\leq O\left(\frac{\sum_{\tau=0}^{t_0-1} \eta_\tau + d \sum_{\tau=t_0}^{t-1} \eta_\tau^{\frac{3}{2}}}{\sum_{\tau=0}^{t-1} \eta_\tau}\right),$$

where the second inequality follows by the assumption that $\eta_t \to 0$. This finishes the proof. $\square$

# 7    Conclusion and Future Work

In this paper, we study the implicit bias of Adam under a challenging but insightful setting where the "stability constant $\epsilon$" is negligible and set to zero. We demonstrate that Adam has an implicit bias converging towards the maximum $\ell_\infty$-margin solution, and such convergence occurs in polynomials of time for a general class of learning rates. This result further helps to understand the distinctions between Adam and (stochastic) gradient descent with/without momentum, whose iterates will eventually converge to the maximum $\ell_2$-margin solution with an $O(1/\log t)$ convergence rate. This finding aligns with the implicit bias of Adam observed in experiments, for both cases the stability constant $\epsilon$ is zero and $10^{-8}$. We predict that similar result can be extended to homogeneous neural networks, and we believe that this is a good future work direction. Moreover, since this paper focuses on full-batch Adam, another feasible future work is to investigate the implicit bias of stochastic Adam based on our results. In addition, establishing matching lower bounds for the loss and margin convergence rates for Adam is also an interesting future work direction.

## Acknowledgments and Disclosure of Funding

We would like to thank the anonymous reviewers and area chairs for their helpful comments. DZ is supported in part by NSFC 62306252, Guangdong NSF 2024A1515012444, Hong Kong ECS award 27309624, and the central fund from HKU IDS. YC is supported in part by NSFC 12301657 and Hong Kong ECS award 27308624.

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

# A Proof in Section 6

## A.1 Preliminary Lemma

It can be figured out that only the product $y_i \cdot \mathbf{x}_i$ is concerned in (3.1). Therefore, we define $\mathbf{z}_i = y_i \cdot \mathbf{x}_i$, then (3.1) could be re-written as

$$\mathcal{R}(\mathbf{w}) = \frac{1}{n} \sum_{i=1}^n \ell(\langle \mathbf{w}, \mathbf{z}_i \rangle).$$

We also define $\mathbf{Z} = [\mathbf{z}_1, \mathbf{z}_2, \cdots, \mathbf{z}_n]^\top \in \mathbb{R}^{n \times d}$ to denote our sample with $i-$th row is $\mathbf{z}_i^\top$, and we will use $\mathbf{Z} \in \mathbb{R}^{n \times d}$ to denote the data sample instead of $\{(\mathbf{x}_i, y_i)\}_{i=1}^n$ in the following paragraphs. Then $\mathcal{G}(\mathbf{w}) = \frac{1}{n} \sum_{i=1}^n -\ell'(\langle \mathbf{w}, \mathbf{z}_i \rangle)$, and we introduce $L'(\mathbf{w}) = [-\frac{1}{n}\ell'(\langle \mathbf{w}, \mathbf{z}_1 \rangle), -\frac{1}{n}\ell'(\langle \mathbf{w}, \mathbf{z}_2 \rangle), \cdots, -\frac{1}{n}\ell'(\langle \mathbf{w}, \mathbf{z}_n \rangle)]^\top$, which means $\mathcal{G}(\mathbf{w}) = \|L'(\mathbf{w})\|_1$. The following lemma reveals the relationship between the maximum margin $\gamma$ and $\nabla\mathcal{R}(\mathbf{w}_t)$ by duality.

**Lemma A.1.** For data sample $\mathbf{Z}$ under Assumption 4.1 and maximum $\ell_\infty$-margin $\gamma$ as defined in (4.1), then

$$\gamma \leq \min_{\mathbf{r} \in \Delta_n} \|\mathbf{Z}^\top \mathbf{r}\|_1,$$

where $\Delta_n = \{\mathbf{r} | \mathbf{r} \in \mathbb{R}^n, \sum_{i=1}^n r_i = 1, r_i \geq 0\}$ is the $n$ dimensional simplex.

**Remark A.2.** Since $\frac{L'(\mathbf{w}_t)}{\mathcal{G}(\mathbf{w}_t)} \in \Delta_n$, and $\nabla\mathcal{R}(\mathbf{w}_t) = \mathbf{Z}^\top L'(\mathbf{w}_t)$, we always have $\gamma\mathcal{G}(\mathbf{w}_t) \leq \|\nabla\mathcal{R}(\mathbf{w}_t)\|_1$, which is the essence for proving the convergence direction of gradient-based algorithms. This result was also proposed in [39, 41, 14, 21, 23].

*Proof of Lemma A.1.* Firstly, we introduce a definition of indicator function $\iota(\cdot)$ as

$$\iota_E(z) = \begin{cases} 0, \text{if } z \in E; \\ +\infty, \text{if } z \notin E, \end{cases}$$

where $E$ is any set. Let $f(\mathbf{r}) = \iota_{\Delta_n}(\mathbf{r})$ and $g(\mathbf{z}) = \|\mathbf{z}\|_1$, then we could derive that $f^*(\mathbf{r}^*) = \max_{i \in [n]} \langle \mathbf{e}_i, \mathbf{r}^* \rangle$ is the dual function of $f(\mathbf{r})$ and $g^*(\mathbf{z}^*) = \iota_{\|\mathbf{z}^*\|_\infty \leq 1}$ is the dual function of $g(\mathbf{z})$. Then by Fenchel-Young inequality, we have

$$\min_{\mathbf{r} \in \Delta_n} \|\mathbf{Z}^\top \mathbf{r}\|_1 = \min_{\mathbf{r} \in \mathbb{R}^n} \left[ f(\mathbf{r}) + g(\mathbf{Z}^\top \mathbf{r}) \right] \geq \max_{\mathbf{w} \in \mathbb{R}^d} \left[ -f^*(\mathbf{Z}\mathbf{w}) - g^*(-\mathbf{w}) \right]$$

$$= \max_{\mathbf{w} \in \mathbb{R}^d} \left[ -\max_{i \in [n]} \mathbf{e}_i^\top \mathbf{Z}\mathbf{w} - \iota_{\|\mathbf{w}\|_\infty \leq 1} \right] = \max_{\|\mathbf{w}\|_\infty \leq 1} \min_{i \in [n]} \mathbf{e}_i^\top \mathbf{Z}\mathbf{w} = \gamma.$$

$\square$

## A.2 Proof of Lemma 6.5

We first introduce the proof of Lemma 6.5 since it will be used for further proof of other lemmas.

*Proof of Lemma 6.5.* By Cauchy-Schwartz inequality, we could derive an upper bound for $\mathbf{m}_t[k]$ as

$$|\mathbf{m}_t[k]| = |\beta_1 \mathbf{m}_{t-1}[k] + (1-\beta_1) \cdot \nabla\mathcal{R}(\mathbf{w}_t)[k]| \leq \sum_{\tau=0}^t \beta_1^\tau (1-\beta_1) \cdot |\nabla\mathcal{R}(\mathbf{w}_{t-\tau})[k]|$$

$$= \sum_{\tau=0}^t \sqrt{\beta_2^\tau(1-\beta_2)} \frac{\beta_1^\tau(1-\beta_1)}{\sqrt{\beta_2^\tau(1-\beta_2)}} \cdot |\nabla\mathcal{R}(\mathbf{w}_{t-\tau})[k]|$$

$$\leq \left( \sum_{\tau=0}^t \beta_2^\tau(1-\beta_2) \cdot \nabla\mathcal{R}(\mathbf{w}_{t-\tau})[k]^2 \right)^{\frac{1}{2}} \left( \sum_{\tau=0}^t \frac{\beta_1^{2\tau}(1-\beta_1)^2}{\beta_2^\tau(1-\beta_2)} \right)^{\frac{1}{2}}$$

$$\leq \alpha\sqrt{\mathbf{v}_t[k]}.$$

The last inequality is from

$$\mathbf{v}_t[k] = \sum_{\tau=0}^{t} \beta_2^{\tau}(1-\beta_2) \cdot \nabla\mathcal{R}(\mathbf{w}_{t-\tau})[k]^2,$$

and

$$\sum_{\tau=0}^{t} \frac{\beta_1^{2\tau}(1-\beta_1)^2}{\beta_2^{\tau}(1-\beta_2)} \leq \frac{(1-\beta_1)^2}{1-\beta_2} \sum_{\tau=0}^{\infty} \left(\frac{\beta_1^2}{\beta_2}\right)^{\tau} = \frac{\beta_2(1-\beta_1)^2}{(1-\beta_2)(\beta_2-\beta_1^2)} = \alpha^2.$$

This finishes the proof. $\qquad\qquad\square$

### A.3 Proof of Lemma 6.1

*Proof of Lemma 6.1.* Let $\alpha = \sqrt{\frac{\beta_2(1-\beta_1)^2}{(1-\beta_2)(\beta_2-\beta_1^2)}}$ be defined in Lemma 6.5. For $\mathbf{m}_t[k]$, it could be rewritten as

$$\mathbf{m}_t[k] = \sum_{\tau=0}^{t} \beta_1^{\tau}(1-\beta_1) \cdot \nabla\mathcal{R}(\mathbf{w}_{t-\tau})[k]$$

$$= (1-\beta_1^{t+1}) \cdot \nabla\mathcal{R}(\mathbf{w}_t)[k] + \sum_{\tau=0}^{t}(1-\beta_1)\beta_1^{\tau} \cdot \left(\nabla\mathcal{R}(\mathbf{w}_{t-\tau})[k] - \nabla\mathcal{R}(\mathbf{w}_t)[k]\right).$$

Therefore the difference between $\mathbf{m}_t[k]$ and $(1-\beta_1^{t+1}) \cdot \nabla\mathcal{R}(\mathbf{w}_t)[k]$ can be bounded as

$$\left|\mathbf{m}_t[k] - (1-\beta_1^{t+1}) \cdot \nabla\mathcal{R}(\mathbf{w}_t)[k]\right| = \left|\sum_{\tau=0}^{t}(1-\beta_1)\beta_1^{\tau} \cdot \left(\nabla\mathcal{R}(\mathbf{w}_{t-\tau})[k] - \nabla\mathcal{R}(\mathbf{w}_t)[k]\right)\right|$$

$$= \left|\sum_{\tau=0}^{t}(1-\beta_1)\beta_1^{\tau}\left(\frac{1}{n}\sum_{i=1}^{n}\left[\ell'(\langle\mathbf{w}_{t-\tau},\mathbf{z}_i\rangle) - \ell'(\langle\mathbf{w}_t,\mathbf{z}_i\rangle)\right] \cdot \mathbf{z}_i[k]\right)\right|$$

$$\leq \sum_{\tau=0}^{t}(1-\beta_1)\beta_1^{\tau}\left(\frac{1}{n}\sum_{i=1}^{n}\left|\ell'(\langle\mathbf{w}_t,\mathbf{z}_i\rangle)\right|\left|\frac{\ell'(\langle\mathbf{w}_{t-\tau},\mathbf{z}_i\rangle)}{\ell'(\langle\mathbf{w}_t,\mathbf{z}_i\rangle)} - 1\right|\left|\mathbf{z}_i[k]\right|\right)$$

$$\leq (1-\beta_1)B\sum_{\tau=0}^{t}\beta_1^{\tau}\left(\frac{1}{n}\sum_{i=1}^{n}\left|\ell'(\langle\mathbf{w}_t,\mathbf{z}_i\rangle)\right|\right)\left(e^{\alpha B\sum_{\tau'=1}^{\tau}\eta_{t-\tau'}} - 1\right)$$

$$\leq (1-\beta_1)Bc_2\eta_t \cdot \mathcal{G}(\mathbf{w}_t) = c_m\eta_t \cdot \mathcal{G}(\mathbf{w}_t).$$

The second inequality holds since $|\mathbf{z}_i[k]| \leq \|\mathbf{z}_i\|_1 \leq B$, and for both $\ell \in \{\ell_{\exp}, \ell_{\log}\}$, we have

$$\left|\frac{\ell'(\langle\mathbf{w}_{t-\tau},\mathbf{z}_i\rangle)}{\ell'(\langle\mathbf{w}_t,\mathbf{z}_i\rangle)} - 1\right| \leq e^{|\langle\mathbf{w}_t-\mathbf{w}_{t-\tau},\mathbf{z}_i\rangle|} - 1 \leq e^{\|\mathbf{w}_t-\mathbf{w}_{t-\tau}\|_\infty\|\mathbf{z}_i\|_1} - 1 \leq e^{\alpha B\sum_{\tau'=1}^{\tau}\eta_{t-\tau'}} - 1,$$

by Lemma 6.5 and Lemma C.5. The last inequality holds since $\sum_{\tau=0}^{t}\beta_1^{\tau}\left(e^{\alpha B\sum_{\tau'=1}^{\tau}\eta_{t-\tau'}} - 1\right) \leq c_2\eta_t$ by our Assumption 4.4. Similarly for $\mathbf{v}_t[k]$, we also have

$$\left|\mathbf{v}_t[k] - (1-\beta_2^{t+1}) \cdot \nabla\mathcal{R}(\mathbf{w}_t)[k]^2\right|$$

$$= \left|\sum_{\tau=0}^{t}(1-\beta_2)\beta_2^{\tau} \cdot \left(\nabla\mathcal{R}(\mathbf{w}_{t-\tau})[k]^2 - \nabla\mathcal{R}(\mathbf{w}_t)[k]^2\right)\right|$$

$$= \left|\sum_{\tau=0}^{t}(1-\beta_2)\beta_2^{\tau}\left(\frac{1}{n^2}\sum_{i,j=1}^{n}\left[\ell'(\langle\mathbf{w}_{t-\tau},\mathbf{z}_i\rangle)\ell'(\langle\mathbf{w}_{t-\tau},\mathbf{z}_j\rangle) - \ell'(\langle\mathbf{w}_t,\mathbf{z}_i\rangle)\ell'(\langle\mathbf{w}_t,\mathbf{z}_j\rangle)\right] \cdot \mathbf{z}_i[k]\mathbf{z}_j[k]\right)\right|$$

$$\leq \left|\sum_{\tau=0}^{t}(1-\beta_2)\beta_2^{\tau}\left(\frac{1}{n^2}\sum_{i,j=1}^{n}\left|\ell'(\langle\mathbf{w}_t,\mathbf{z}_i\rangle)\right|\left|\ell'(\langle\mathbf{w}_t,\mathbf{z}_j\rangle)\right|\left|\frac{\ell'(\langle\mathbf{w}_{t-\tau},\mathbf{z}_i\rangle)\ell'(\langle\mathbf{w}_{t-\tau},\mathbf{z}_j\rangle)}{\ell'(\langle\mathbf{w}_t,\mathbf{z}_i\rangle)\ell'(\langle\mathbf{w}_t,\mathbf{z}_j\rangle)} - 1\right|\left|\mathbf{z}_i[k]\right|\left|\mathbf{z}_j[k]\right|\right)\right|$$

$$\leq 3(1-\beta_2)B^2 \sum_{\tau=0}^{t} \beta_2^\tau \left( \frac{1}{n^2} \sum_{i,j=1}^{n} |\ell'(\langle \mathbf{w}_t, \mathbf{z}_i \rangle)||\ell'(\langle \mathbf{w}_t, \mathbf{z}_j \rangle)| \right) \left( e^{2\alpha B \sum_{\tau'=1}^{\tau} \eta_{t-\tau'}} - 1 \right)$$

$$\leq 3(1-\beta_2)B^2 c_2 \eta_t \cdot \mathcal{G}(\mathbf{w}_t)^2 = c_v^2 \eta_t \cdot \mathcal{G}(\mathbf{w}_t)^2.$$

Similarly, the second inequality holds since $|\mathbf{z}_i[k]||\mathbf{z}_j[k]| \leq \|\mathbf{z}_i\|_1 \|\mathbf{z}_j\|_1 \leq B^2$, and for both $\ell \in \{\ell_{\exp}, \ell_{\log}\}$, we have

$$\left| \frac{\ell'(\langle \mathbf{w}_{t-\tau}, \mathbf{z}_i \rangle)\ell'(\langle \mathbf{w}_{t-\tau}, \mathbf{z}_j \rangle)}{\ell'(\langle \mathbf{w}_t, \mathbf{z}_i \rangle)\ell'(\langle \mathbf{w}_t, \mathbf{z}_j \rangle)} - 1 \right|$$

$$\leq \left( e^{|\langle \mathbf{w}_t - \mathbf{w}_{t-\tau}, \mathbf{z}_i \rangle|} - 1 \right) + \left( e^{|\langle \mathbf{w}_t - \mathbf{w}_{t-\tau}, \mathbf{z}_j \rangle|} - 1 \right) + \left( e^{|\langle \mathbf{w}_t - \mathbf{w}_{t-\tau}, \mathbf{z}_i + \mathbf{z}_j \rangle|} - 1 \right)$$

$$\leq \left( e^{\|\mathbf{w}_t - \mathbf{w}_{t-\tau}\|_\infty \|\mathbf{z}_i\|_1} - 1 \right) + \left( e^{\|\mathbf{w}_t - \mathbf{w}_{t-\tau}\|_\infty \|\mathbf{z}_j\|_1} - 1 \right) + \left( e^{\|\mathbf{w}_t - \mathbf{w}_{t-\tau}\|_\infty \|\mathbf{z}_i + \mathbf{z}_j\|_1} - 1 \right)$$

$$\leq 3 \left( e^{2\alpha B \sum_{\tau'=1}^{\tau} \eta_{t-\tau'}} - 1 \right)$$

by Lemma 6.5 and Lemma C.6. The last inequality holds since $\sum_{\tau=0}^{t} \beta_2^\tau \left( e^{2\alpha B \sum_{\tau'=1}^{\tau} \eta_{t-\tau'}} - 1 \right) \leq c_2 \eta_t$ by our Assumption 4.4. Now, it remains to show the upper bound for $\left| \sqrt{\mathbf{v}_t[k]} - \sqrt{1-\beta_2^{t+1}} \cdot |\nabla \mathcal{R}(\mathbf{w}_t)[k]| \right|$. Notice that both $\mathbf{v}_t[k]$ and $(1-\beta_2^{t+1}) \cdot \nabla \mathcal{R}(\mathbf{w}_t)[k]^2$ are positive and for two positive numbers $a$ and $b$, $|a^2 - b^2| = |a-b||a+b| \geq |a-b|^2$, therefore we finally conclude that,

$$\left| \sqrt{\mathbf{v}_t[k]} - \sqrt{1-\beta_2^{t+1}} \cdot |\nabla \mathcal{R}(\mathbf{w}_t)[k]| \right| \leq c_v \sqrt{\eta_t} \cdot \mathcal{G}(\mathbf{w}_t).$$

This finishes the proof. $\qquad \square$

### A.4 Proof of Lemma 6.2

Before we prove Lemma 6.2, we first introduce and prove Lemma A.3, which will be used for proving Lemma 6.2.

**Lemma A.3.** Under the same condition in Theorem 4.5, there exists $t_1 = t_1(\beta_1, \beta_2, \gamma)$, such that when $t > t_1$, we have

$$\left| \left\langle \nabla \mathcal{R}(\mathbf{w}_t), \frac{\mathbf{m}_t}{\sqrt{\mathbf{v}_t}} \right\rangle - \left\| \nabla \mathcal{R}(\mathbf{w}_t) \right\|_1 \right| \leq 4 \sqrt{\frac{\beta_1^{t+1}}{1-\beta_2^{t+1}}} \cdot \left\| \nabla \mathcal{R}(\mathbf{w}_t) \right\|_1$$

$$+ \frac{d}{\sqrt{1-\beta_2}} \left( \frac{6c_v}{\sqrt{1-\beta_2^{t+1}}} \sqrt{\eta_t} + 3c_m \eta_t \right) \cdot \mathcal{G}(\mathbf{w}_t), \quad \text{(A.1)}$$

where $c_m$ and $c_v$ are both constants which only depend on $\beta_1$, $\beta_2$ and $B$.

*Proof of Lemma A.3.* By Lemma 6.1, we could re-write $\mathbf{m}_t[k]$ and $\sqrt{\mathbf{v}_t[k]}$ as

$$\mathbf{m}_t[k] = (1-\beta_1^{t+1}) \cdot \nabla \mathcal{R}(\mathbf{w}_t)[k] + c_m \eta_t \cdot \mathcal{G}(\mathbf{w}_t) \cdot \epsilon_{t,m,k},$$

and

$$\sqrt{\mathbf{v}_t[k]} = \sqrt{1-\beta_2^{t+1}} \cdot \left| \nabla \mathcal{R}(\mathbf{w}_t)[k] \right| + c_v \sqrt{\eta_t} \cdot \mathcal{G}(\mathbf{w}_t) \cdot \epsilon_{t,v,k} > 0,$$

where $\epsilon_{t,m,k}$ and $\epsilon_{t,v,k}$ are just some error terms with $|\epsilon_{t,m,k}|, |\epsilon_{t,v,k}| \leq 1$. Then we can calculate the inner-product $\langle \nabla \mathcal{R}(\mathbf{w}_t), \frac{\mathbf{m}_t}{\sqrt{\mathbf{v}_t}} \rangle$ for each iteration as

$$\left\langle \nabla \mathcal{R}(\mathbf{w}_t), \frac{\mathbf{m}_t}{\sqrt{\mathbf{v}_t}} \right\rangle = \left\| \nabla \mathcal{R}(\mathbf{w}_t) \right\|_1 + \underbrace{\sum_{k=1}^{d} \nabla \mathcal{R}(\mathbf{w}_t)[k] \cdot \left( \frac{\mathbf{m}_t[k]}{\sqrt{\mathbf{v}_t[k]}} - \frac{\nabla \mathcal{R}(\mathbf{w}_t)[k]}{|\nabla \mathcal{R}(\mathbf{w}_t)[k]|} \right)}_{(*)} \cdot$$

Moreover, we let

$$\xi_{t,k} = \nabla\mathcal{R}(\mathbf{w}_t)[k] \cdot \left( \frac{\mathbf{m}_t[k]}{\sqrt{\mathbf{v}_t[k]}} - \frac{\nabla\mathcal{R}(\mathbf{w}_t)[k]}{|\nabla\mathcal{R}(\mathbf{w}_t)[k]|} \right)$$

$$= \nabla\mathcal{R}(\mathbf{w}_t)[k] \cdot \left( \frac{(1-\beta_1^{t+1}) \cdot \nabla\mathcal{R}(\mathbf{w}_t)[k] + c_m \eta_t \cdot \mathcal{G}(\mathbf{w}_t) \cdot \epsilon_{t,m,k}}{\sqrt{1-\beta_2^{t+1}} \cdot |\nabla\mathcal{R}(\mathbf{w}_t)[k]| + c_v\sqrt{\eta_t} \cdot \mathcal{G}(\mathbf{w}_t) \cdot \epsilon_{t,v,k}} - \frac{\nabla\mathcal{R}(\mathbf{w}_t)[k]}{|\nabla\mathcal{R}(\mathbf{w}_t)[k]|} \right),$$

and consider to separate $\xi_{t,k}$ into two complementary parts. The first part is $\xi_{t,k}\mathbb{1}_{A_{t,k}}$, where $A_{t,k} = \left\{ \sqrt{1-\beta_2^{t+1}} \cdot |\nabla\mathcal{R}(\mathbf{w}_t)[k]| \geq 2c_v\sqrt{\eta_t} \cdot \mathcal{G}(\mathbf{w}_t) \cdot |\epsilon_{t,v,k}| \right\}$. While another part is $\xi_{t,k}\mathbb{1}_{A_{t,k}^c}$, where $A_{t,k}^c = \left\{ \sqrt{1-\beta_2^{t+1}} \cdot |\nabla\mathcal{R}(\mathbf{w}_t)[k]| < 2c_v\sqrt{\eta_t} \cdot \mathcal{G}(\mathbf{w}_t) \cdot |\epsilon_{t,v,k}| \right\}$. By such separation, we have

$$|(*)| = \left| \sum_{k=1}^d \xi_{t,k}\mathbb{1}_{A_{t,k}} + \xi_{t,k}\mathbb{1}_{A_{t,k}^c} \right| \leq \left| \sum_{k=1}^d \xi_{t,k}\mathbb{1}_{A_{t,k}} \right| + \left| \sum_{k=1}^d \xi_{t,k}\mathbb{1}_{A_{t,k}^c} \right|$$

We calculate it part by part. For the first part $\left| \sum_{k=1}^d \xi_{t,k}\mathbb{1}_{A_{t,k}} \right|$, we have

$$\left| \sum_{k=1}^d \xi_{t,k}\mathbb{1}_{A_{t,k}} \right| \leq \sum_{k=1}^d \frac{\left| 1 - \beta^{t+1} - \sqrt{1-\beta_2^{t+1}} \right| \cdot |\nabla\mathcal{R}(\mathbf{w}_t)[k]|^3 + \left(c_m\eta_t + c_v\sqrt{\eta_t}\right) \cdot \mathcal{G}(\mathbf{w}_t) \cdot |\nabla\mathcal{R}(\mathbf{w}_t)[k]|^2}{\left( \sqrt{1-\beta_2^{t+1}} \cdot |\nabla\mathcal{R}(\mathbf{w}_t)[k]| + c_v\sqrt{\eta_t} \cdot \mathcal{G}(\mathbf{w}_t) \cdot \epsilon_{t,v,k} \right) \cdot |\nabla\mathcal{R}(\mathbf{w}_t)[k]|} \mathbb{1}_{A_{t,k}}$$

$$\leq \sum_{k=1}^d \frac{\left| 1 - \beta^{t+1} - \sqrt{1-\beta_2^{t+1}} \right| \cdot |\nabla\mathcal{R}(\mathbf{w}_t)[k]|^3 + \left(c_m\eta_t + c_v\sqrt{\eta_t}\right) \cdot \mathcal{G}(\mathbf{w}_t) \cdot |\nabla\mathcal{R}(\mathbf{w}_t)[k]|^2}{\frac{1}{2}\sqrt{1-\beta_2^{t+1}} \cdot |\nabla\mathcal{R}(\mathbf{w}_t)[k]|^2}$$

$$\leq 4\sqrt{\frac{\beta_1^{t+1}}{1-\beta_2^{t+1}}} \cdot \left\| \nabla\mathcal{R}(\mathbf{w}_t) \right\|_1 + \frac{2d}{\sqrt{1-\beta_2^{t+1}}} \left( c_m\eta_t + c_v\sqrt{\eta_t} \right) \cdot \mathcal{G}(\mathbf{w}_t). \qquad \text{(A.2)}$$

The first inequality is derived by triangle inequality and $|\epsilon_{t,m,k}|, |\epsilon_{t,v,k}| \leq 1$. The second inequality holds since $\sqrt{1-\beta_2^{t+1}} \cdot |\nabla\mathcal{R}(\mathbf{w}_t)[k]| + c_v\sqrt{\eta_t} \cdot \mathcal{G}(\mathbf{w}_t) \cdot \epsilon_{t,v,k} > \frac{1}{2}\sqrt{1-\beta_2^{t+1}} \cdot |\nabla\mathcal{R}(\mathbf{w}_t)[k]|$ when $\mathbb{1}_{A_{t,k}} = 1$. And the last inequality is simply due to an arithmetic result that

$$\left| 1 - \beta_1^{t+1} - \sqrt{1-\beta_2^{t+1}} \right| \leq \left| 1 - \sqrt{1-\beta_2^{t+1}} \right| + \beta_1^{t+1} \leq \sqrt{1-1+\beta_2^{t+1}} + \beta_1^{t+1} \leq 2\beta_1^{\frac{t+1}{2}}.$$

Then for another part $\left| \sum_{k=1}^d \xi_{t,k}\mathbb{1}_{A_{t,k}^c} \right|$, we use the property $\sqrt{\mathbf{v}_t[k]} \geq \sqrt{1-\beta_2} \cdot |\nabla\mathcal{R}(\mathbf{w}_t)[k]|$ to derive an upper bound as

$$\left| \sum_{k=1}^d \xi_{t,k}\mathbb{1}_{A_{t,k}^c} \right| \leq \sum_{k=1}^d |\nabla\mathcal{R}(\mathbf{w}_t)[k]| \cdot \left( \frac{|\nabla\mathcal{R}(\mathbf{w}_t)[k]| + c_m\eta_t \cdot \mathcal{G}(\mathbf{w}_t)}{\sqrt{1-\beta_2} \cdot |\nabla\mathcal{R}(\mathbf{w}_t)[k]|} + 1 \right) \mathbb{1}_{A_{t,k}^c}$$

$$\leq \frac{d}{\sqrt{1-\beta_2}} \left( \frac{4c_v}{\sqrt{1-\beta_2^{t+1}}} \sqrt{\eta_t} + c_m\eta_t \right) \cdot \mathcal{R}(\mathbf{w}_t). \qquad \text{(A.3)}$$

The first inequality is derived by triangle inequality and $|\epsilon_{t,m,k}| \leq 1$. The second inequality holds since $|\nabla\mathcal{R}(\mathbf{w}_t)[k]| \leq \frac{2c_v}{\sqrt{1-\beta_2^{t+1}}}\sqrt{\eta_t} \cdot \mathcal{G}(\mathbf{w}_t)$ when $\mathbb{1}_{A_{t,k}^c} = 1$. Combining the results of (A.2),(A.3) and Lemma C.2, we finally prove finish the proof. $\qquad \square$

Now, we are ready to prove Lemma 6.2.

*Proof of Lemma 6.2.* We upper bound $\mathcal{R}(\mathbf{w}_{t+1})$ for $t > t_1$ by second-order Taylor expansion as

$$\mathcal{R}(\mathbf{w}_{t+1}) = \mathcal{R}(\mathbf{w}_t) + \left\langle \nabla\mathcal{R}(\mathbf{w}_t), \mathbf{w}_{t+1} - \mathbf{w}_t \right\rangle + \frac{1}{2}\left(\mathbf{w}_{t+1} - \mathbf{w}_t\right)^\top \nabla^2\mathcal{R}\left(\mathbf{w}_t + \zeta(\mathbf{w}_{t+1} - \mathbf{w}_t)\right)\left(\mathbf{w}_{t+1} - \mathbf{w}_t\right)$$

$$= \mathcal{R}(\mathbf{w}_t) - \left\langle \nabla\mathcal{R}(\mathbf{w}_t), \eta_t \frac{\mathbf{m}_t}{\sqrt{\mathbf{v}_t}} \right\rangle + \frac{1}{2}\left(\eta_t \frac{\mathbf{m}_t}{\sqrt{\mathbf{v}_t}}\right)^\top \nabla^2\mathcal{R}\big(\mathbf{w}_t + \zeta(\mathbf{w}_{t+1} - \mathbf{w}_t)\big)\left(\eta_t \frac{\mathbf{m}_t}{\sqrt{\mathbf{v}_t}}\right)$$

$$\leq \mathcal{R}(\mathbf{w}_t) - \eta_t\left(1 - 4\sqrt{\frac{\beta_1^{t+1}}{1-\beta_2^{t+1}}}\right)\cdot \left\|\nabla\mathcal{R}(\mathbf{w}_t)\right\|_1$$

$$+ \eta_t \frac{d}{\sqrt{1-\beta_2}}\left(\frac{6c_v}{\sqrt{1-\beta_2^{t+1}}}\sqrt{\eta_t} + 3c_m\eta_t\right)\cdot\mathcal{G}(\mathbf{w}_t) + \frac{\eta_t^2\alpha^2 B^2}{2}\cdot\max\left\{\mathcal{G}(\mathbf{w}_t),\mathcal{G}(\mathbf{w}_{t+1})\right\}$$

$$\leq \mathcal{R}(\mathbf{w}_t) - \eta_t\gamma\left(1 - 4\sqrt{\frac{\beta_1^{t+1}}{1-\beta_2^{t+1}}}\right)\cdot\mathcal{G}(\mathbf{w}_t) + \eta_t^{\frac{3}{2}}\frac{6c_v d}{\sqrt{(1-\beta_2)(1-\beta_2^{t+1})}}\cdot\mathcal{G}(\mathbf{w}_t)$$

$$+ \eta_t^2\left(\frac{\alpha^2 B^2 e^{\alpha B\eta_0}}{2} + \frac{3c_m d}{\sqrt{1-\beta_2}}\right)\cdot\mathcal{G}(\mathbf{w}_t).$$

The first inequality is from Lemma A.3, and for the vector $\frac{\mathbf{m}_t}{\sqrt{\mathbf{v}_t}}$,

$$\left(\frac{\mathbf{m}_t}{\sqrt{\mathbf{v}_t}}\right)^\top \nabla^2\mathcal{R}(\mathbf{w})\left(\frac{\mathbf{m}_t}{\sqrt{\mathbf{v}_t}}\right) \leq \frac{1}{n}\sum_{i=1}^n \ell''(\langle\mathbf{w},\mathbf{z}_i\rangle)\left\|\frac{\mathbf{m}_t}{\sqrt{\mathbf{v}_t}}\right\|_\infty^2 \|\mathbf{z}_i\|_1^2 \leq \alpha^2 B^2\cdot\mathcal{G}(\mathbf{w})$$

by Lemma C.2 and Lemma 6.5, and $\mathcal{G}\big(\mathbf{w}_t + \zeta(\mathbf{w}_{t+1} - \mathbf{w}_t)\big) \leq \max\left\{\mathcal{G}(\mathbf{w}_t),\mathcal{G}(\mathbf{w}_{t+1})\right\}$ from convexity of $\mathcal{G}(\mathbf{w})$. The last inequality is from $\frac{\mathcal{G}(\mathbf{w}_{t+1})}{\mathcal{G}(\mathbf{w}_t)} \leq e^{\alpha B\eta_t} \leq e^{\alpha B\eta_0}$ by Lemma C.5 and $\gamma\mathcal{G}(\mathbf{w}_t) \leq \|\nabla\mathcal{R}(\mathbf{w}_t)\|_1$ by Lemma A.1. $\qquad\square$

## A.5 Proof of Lemma 6.3

*Proof of Lemma 6.3.* By Lemma 6.2 and Lemma C.2 , we have

$$\mathcal{R}(\mathbf{w}_{t+1}) \leq \mathcal{R}(\mathbf{w}_t)\cdot\left(1 - \gamma\eta_t\cdot\frac{\mathcal{G}(\mathbf{w}_t)}{\mathcal{R}(\mathbf{w}_t)} + \left(C_1\gamma\eta_t\beta_1^{t/2} + \eta_t^{\frac{3}{2}}C_2 d + \eta_t^2 C_2 d\right)\cdot\frac{\mathcal{G}(\mathbf{w}_t)}{\mathcal{R}(\mathbf{w}_t)}\right)$$

$$\leq \mathcal{R}(\mathbf{w}_t)\cdot\exp\left(-\gamma\eta_t\cdot\frac{\mathcal{G}(\mathbf{w}_t)}{\mathcal{R}(\mathbf{w}_t)} + C_1\gamma\eta_t\beta_1^{t/2} + C_2 d\cdot(\eta_t^{\frac{3}{2}} + \eta_t^2)\right)$$

$$\leq \frac{\log 2}{n}\cdot\exp\left(-\gamma\sum_{\tau=t_0}^t \eta_\tau\cdot\frac{\mathcal{G}(\mathbf{w}_\tau)}{\mathcal{R}(\mathbf{w}_\tau)} + C_2 d\left(\sum_{\tau=t_0}^t \eta_\tau^{\frac{3}{2}} + \sum_{\tau=t_0}^t \eta_\tau^2\right) + \frac{C_1\gamma\eta_{t_0}\beta_1^{\frac{t_0+1}{2}}}{1-\sqrt{\beta_1}}\right).$$

$$\tag{A.4}$$

for all $t \geq t_0$. By Lemma C.4, we can derive that $\langle\mathbf{w}_t,\mathbf{z}_i\rangle \geq 0$ for all $i \in [n]$ and $t \geq t_0$. Then we have $e^{-\min_{i\in[n]}\langle\mathbf{w}_t,\mathbf{z}_i\rangle} \leq \frac{1}{\log 2}\max_{i\in[n]}\ell(\langle\mathbf{w}_t,\mathbf{z}_i\rangle) \leq \frac{n}{\log 2}\mathcal{R}(\mathbf{w}_t)$ by Lemma C.3. Plugging this result into (A.4) and taking log on both sides, we finish the proof for Lemma 6.3. $\qquad\square$

## A.6 Proof of Lemma 6.4

Before we prove Lemma 6.4, we first present and prove Lemma A.4 which will be used for proving Lemma 6.4.

**Lemma A.4.** For Adam iterations defined in (3.2)-(3.4) with $\beta_1 \leq \beta_2$, for any $t_0 \in \mathbb{N}_+$, $t > t_0$, and all $k \in [d]$,

$$\big|\mathbf{w}_t[k] - \mathbf{w}_{t_0}[k]\big| \leq \left(\sum_{\tau=t_0}^{t-1} \eta_\tau\right)\cdot$$

$$\left(1 + \frac{\beta_2 - \beta_1}{1-\beta_2}\frac{\sum_{\tau=t_0}^{t-1}\beta_1^{\tau-t_0}\eta_\tau}{\sum_{\tau=t_0}^{t-1}\eta_\tau} + \frac{(\beta_2-\beta_1)(1-\beta_1)}{1-\beta_2}\sum_{\tau=t_0}^{t-1}\frac{\eta_\tau\frac{1-\beta_1^{\tau-t_0}}{1-\beta_1} - \sum_{\tau'=1}^{t-\tau-1}\eta_{\tau+\tau'}\beta_1^{\tau'-1}}{\sum_{\tau=t_0}^{t-1}\eta_\tau}\log(\mathbf{v}_\tau[k])\right)^{\frac{1}{2}}.$$

$$\tag{A.5}$$

*Proof of Lemma A.4.* If we consider implementing the Cauchy-Schwartz inequality on the sum of the iterations, we can get,

$$\left|\mathbf{w}_t[k] - \mathbf{w}_{t_0}[k]\right| = \left|\sum_{\tau=t_0}^{t-1} \eta_\tau \frac{\mathbf{m}_\tau[k]}{\sqrt{\mathbf{v}_\tau[k]}}\right|$$

$$= \left|\sum_{\tau=t_0}^{t-1} \frac{\eta_\tau}{\sqrt{\mathbf{v}_\tau[k]}}\left(\beta_1^{\tau-t_0+1}\mathbf{m}_{t_0-1}[k] + \sum_{\tau'=0}^{\tau-t_0} \beta_1^{\tau'}(1-\beta_1)\cdot\nabla\mathcal{R}(\mathbf{w}_{\tau-\tau'})[k]\right)\right|$$

$$\leq \left[\sum_{\tau=t_0}^{t-1} \frac{\eta_\tau}{\mathbf{v}_\tau[k]}\left(\beta_1^{\tau-t_0+1}\mathbf{m}_{t_0-1}[k]^2 + \sum_{\tau'=0}^{\tau-t_0} \beta_1^{\tau'}(1-\beta_1)\cdot\nabla\mathcal{R}(\mathbf{w}_{\tau-\tau'})[k]^2\right)\right]^{\frac{1}{2}}$$

$$\cdot \left[\sum_{\tau=t_0}^{t-1} \eta_\tau\left(\beta_1^{\tau-t_0+1} + \sum_{\tau'=0}^{\tau-t_0} \beta_1^{\tau'}(1-\beta_1)\right)\right]^{\frac{1}{2}}$$

$$= \underbrace{\left[\sum_{\tau=t_0}^{t-1}\left(\frac{\eta_\tau}{\mathbf{v}_\tau[k]}\beta_1^{\tau-t_0+1}\mathbf{m}_{t_0-1}[k]^2 + \sum_{\tau'=0}^{\tau-t_0}\eta_\tau\beta_1^{\tau'}\frac{1-\beta_1}{1-\beta_2}\frac{\mathbf{v}_{\tau-\tau'}[k]-\beta_2\mathbf{v}_{\tau-\tau'-1}[k]}{\mathbf{v}_\tau[k]}\right)\right]^{\frac{1}{2}}}_{(*)}\left(\sum_{\tau=t_0}^{t-1}\eta_\tau\right)^{\frac{1}{2}}.$$

(A.6)

The inequality is by Cauchy-Schwartz inequality and the second equality is from

$$(1-\beta_2)\cdot\nabla\mathcal{R}(\mathbf{w}_{\tau-\tau'})[k]^2 = \mathbf{v}_{\tau-\tau'}[k] - \beta_2\mathbf{v}_{\tau-\tau'-1}[k],$$

and

$$\sum_{\tau=t_0}^{t-1}\eta_\tau\left(\beta_1^{\tau-t_0+1} + \sum_{\tau'=0}^{\tau-t_0}\beta_1^{\tau'}(1-\beta_1)\right) = \sum_{\tau=t_0}^{t-1}\eta_\tau\left(\beta_1^{\tau-t_0+1} + 1 - \beta_1^{\tau-t_0+1}\right) = \sum_{\tau=t_0}^{t-1}\eta_\tau.$$

For the first part $(*)$ defined in (A.6), we could re-arrange it as,

$$(*) = \sum_{\tau=t_0}^{t-1}\frac{\eta_\tau}{\mathbf{v}_\tau[k]}\beta_1^{\tau-t_0+1}\mathbf{m}_{t_0-1}[k]^2$$

$$+ \sum_{\tau=t_0}^{t-1}\eta_\tau(1-\beta_1)\frac{\mathbf{v}_\tau[k]-\beta_2\beta_1^{\tau-t_0}\mathbf{v}_{t_0-1}[k]+(\beta_1-\beta_2)\sum_{\tau'=1}^{\tau-t_0}\beta_1^{\tau'-1}\mathbf{v}_{\tau-\tau'}[k]}{(1-\beta_2)\mathbf{v}_\tau[k]}$$

$$\leq \sum_{\tau=t_0}^{t-1}\frac{\eta_\tau}{\mathbf{v}_\tau[k]}\beta_1^{\tau-t_0}\left(\beta_1\alpha^2 - \frac{\beta_2(1-\beta_1)}{1-\beta_2}\right)\mathbf{v}_{t_0-1}[k] + \frac{1-\beta_1}{1-\beta_2}\sum_{\tau=t_0}^{t-1}\eta_\tau$$

$$+ \frac{(1-\beta_1)(\beta_1-\beta_2)}{1-\beta_2}\sum_{\tau=t_0}^{t-1}\sum_{\tau'=1}^{\tau-t_0}\eta_\tau\beta_1^{\tau'-1}\frac{\mathbf{v}_{\tau-\tau'}[k]}{\mathbf{v}_\tau[k]}$$

$$\leq \frac{1-\beta_1}{1-\beta_2}\sum_{\tau=t_0}^{t-1}\eta_\tau + \frac{(1-\beta_1)(\beta_1-\beta_2)}{1-\beta_2}\sum_{\tau=t_0}^{t-1}\sum_{\tau'=1}^{\tau-t_0}\eta_\tau\beta_1^{\tau'-1}\frac{\mathbf{v}_{\tau-\tau'}[k]}{\mathbf{v}_\tau[k]}$$

$$= \sum_{\tau=t_0}^{t-1}\eta_\tau + \frac{\beta_2-\beta_1}{1-\beta_2}\sum_{\tau=t_0}^{t-1}\eta_\tau - \frac{(1-\beta_1)(\beta_2-\beta_1)}{1-\beta_2}\sum_{\tau=t_0}^{t-1}\sum_{\tau'=1}^{\tau-t_0}\eta_\tau\beta_1^{\tau'-1}\frac{\mathbf{v}_{\tau-\tau'}[k]}{\mathbf{v}_\tau[k]}$$

$$= \sum_{\tau=t_0}^{t-1}\eta_\tau + \frac{\beta_2-\beta_1}{1-\beta_2}\sum_{\tau=t_0}^{t-1}\eta_\tau\left(1 - (1-\beta_1)\sum_{\tau'=1}^{\tau-t_0}\beta_1^{\tau'-1}\frac{\mathbf{v}_{\tau-\tau'}[k]}{\mathbf{v}_\tau[k]}\right)$$

$$= \sum_{\tau=t_0}^{t-1}\eta_\tau + \frac{\beta_2-\beta_1}{1-\beta_2}\sum_{\tau=t_0}^{t-1}\eta_\tau\left(\beta_1^{\tau-t_0} + (1-\beta_1)\sum_{\tau'=1}^{\tau-t_0}\beta_1^{\tau'-1} - (1-\beta_1)\sum_{\tau'=1}^{\tau-t_0}\beta_1^{\tau'-1}\frac{\mathbf{v}_{\tau-\tau'}[k]}{\mathbf{v}_\tau[k]}\right)$$

$$= \sum_{\tau=t_0}^{t-1}\eta_\tau + \frac{\beta_2-\beta_1}{1-\beta_2}\sum_{\tau=t_0}^{t-1}\eta_\tau\left(\beta_1^{\tau-t_0} + (1-\beta_1)\sum_{\tau'=1}^{\tau-t_0}\beta_1^{\tau'-1}\left(1 - \frac{\mathbf{v}_{\tau-\tau'}[k]}{\mathbf{v}_\tau[k]}\right)\right)$$

$$\leq \sum_{\tau=t_0}^{t-1} \eta_\tau + \frac{\beta_2 - \beta_1}{1 - \beta_2} \sum_{\tau=t_0}^{t-1} \eta_\tau \left( \beta_1^{\tau - t_0} + (1 - \beta_1) \sum_{\tau'=1}^{\tau - t_0} \beta_1^{\tau'-1} \log \left( \frac{\mathbf{v}_\tau[k]}{\mathbf{v}_{\tau-\tau'}[k]} \right) \right)$$

$$= \sum_{\tau=t_0}^{t-1} \eta_\tau + \frac{\beta_2 - \beta_1}{1 - \beta_2} \sum_{\tau=t_0}^{t-1} \eta_\tau \beta_1^{\tau - t_0} + \frac{(\beta_2 - \beta_1)(1 - \beta_1)}{1 - \beta_2} \sum_{\tau=t_0}^{t-1} \eta_\tau \sum_{\tau'=1}^{\tau - t_0} \beta_1^{\tau'-1} \left[ \log(\mathbf{v}_\tau[k]) - \log(\mathbf{v}_{\tau-\tau'}[k]) \right]$$

$$= \sum_{\tau=t_0}^{t-1} \eta_\tau + \frac{\beta_2 - \beta_1}{1 - \beta_2} \sum_{\tau=t_0}^{t-1} \eta_\tau \beta_1^{\tau - t_0}$$

$$+ \frac{(\beta_2 - \beta_1)(1 - \beta_1)}{1 - \beta_2} \left( \sum_{\tau=t_0}^{t-1} \eta_\tau \frac{1 - \beta_1^{\tau - t_0}}{1 - \beta_1} \log(\mathbf{v}_\tau[k]) - \sum_{\tau=t_0}^{t-1} \eta_\tau \sum_{\tau'=1}^{\tau - t_0} \beta_1^{\tau'-1} \log(\mathbf{v}_{\tau-\tau'}[k]) \right)$$

$$\overset{\tau^* = \tau - \tau'}{=\!=\!=\!=\!=} \sum_{\tau=t_0}^{t-1} \eta_\tau + \frac{\beta_2 - \beta_1}{1 - \beta_2} \sum_{\tau=t_0}^{t-1} \eta_\tau \beta_1^{\tau - t_0}$$

$$+ \frac{(\beta_2 - \beta_1)(1 - \beta_1)}{1 - \beta_2} \left( \sum_{\tau=t_0}^{t-1} \eta_\tau \frac{1 - \beta_1^{\tau - t_0}}{1 - \beta_1} \log(\mathbf{v}_\tau[k]) - \sum_{\tau=t_0}^{t-1} \eta_\tau \sum_{\tau^*=t_0}^{\tau-1} \beta_1^{\tau - \tau^* -1} \log(\mathbf{v}_{\tau^*}[k]) \right)$$

$$= \sum_{\tau=t_0}^{t-1} \eta_\tau + \frac{\beta_2 - \beta_1}{1 - \beta_2} \sum_{\tau=t_0}^{t-1} \eta_\tau \beta_1^{\tau - t_0}$$

$$+ \frac{(\beta_2 - \beta_1)(1 - \beta_1)}{1 - \beta_2} \left( \sum_{\tau=t_0}^{t-1} \eta_\tau \frac{1 - \beta_1^{\tau - t_0}}{1 - \beta_1} \log(\mathbf{v}_\tau[k]) - \sum_{\tau^*=t_0}^{t-1} \log(\mathbf{v}_{\tau^*}[k]) \sum_{\tau=\tau^*+1}^{t-1} \eta_\tau \beta_1^{\tau - \tau^* -1} \right)$$

$$= \sum_{\tau=t_0}^{t-1} \eta_\tau + \frac{\beta_2 - \beta_1}{1 - \beta_2} \sum_{\tau=t_0}^{t-1} \eta_\tau \beta_1^{\tau - t_0}$$

$$+ \frac{(\beta_2 - \beta_1)(1 - \beta_1)}{1 - \beta_2} \sum_{\tau=t_0}^{t-1} \left( \eta_\tau \frac{1 - \beta_1^{\tau - t_0}}{1 - \beta_1} - \sum_{\tau'=1}^{t-\tau-1} \eta_{\tau+\tau'} \beta_1^{\tau'-1} \right) \log(\mathbf{v}_\tau[k]). \quad \text{(A.7)}$$

Plugging (A.7) into (A.6), then we derive the result of (A.5). □

Now we are ready to prove Lemma 6.4.

*Proof of Lemma 6.4.* Considering the last two terms on the RHS of (A.7), for the second term, we can upper-bound it as,

$$\frac{\beta_2 - \beta_1}{1 - \beta_2} \sum_{\tau=t_0}^{t-1} \eta_\tau \beta_1^{\tau - t_0} \leq \frac{\eta_{t_0}(\beta_2 - \beta_1)}{(1 - \beta_1)(1 - \beta_2)},$$

since $\eta_t$ is decreasing. Let $C_5 = c_v^2$ in statement of Lemma A.4. Then for the third term, by Lemma 6.1 and our condition $\mathcal{R}(\mathbf{w}_t) \leq \frac{1}{\sqrt{B^2 + C_5 \eta_0}}$, we have

$$\mathbf{v}_t[k] \leq \nabla \mathcal{R}(\mathbf{w}_t)[k]^2 + c_v^2 \eta_t \cdot \mathcal{G}(\mathbf{w}_t)^2 \leq (B^2 + C_5 \eta_0) \cdot \mathcal{R}(\mathbf{w}_t)^2 \leq 1$$

for all $k \in [d]$, which implies that $\log(\mathbf{v}_t[k]) < 0$ for all $t > t_0$. On the other hand,

$$\log(\mathbf{v}_t[k]) \geq \log(\beta_2^t (1 - \beta_2) \nabla \mathcal{R}(\mathbf{w}_0)[k]^2) \geq t \log \beta_2 + \log(1 - \beta_2) + \log \rho,$$

and

$$\eta_\tau \frac{1 - \beta_1^{\tau - t_0}}{1 - \beta_1} - \sum_{\tau'=1}^{t-\tau-1} \eta_{\tau+\tau'} \beta_1^{\tau'-1} \geq \eta_\tau \left( \frac{1 - \beta_1^{\tau - t_0}}{1 - \beta_1} - \sum_{\tau'=1}^{t-\tau-1} \beta_1^{\tau'-1} \right) \geq -\eta_\tau \frac{\beta_1^{\tau - t_0}}{1 - \beta_1}.$$

Combining these results, we can upper-bound the third term as,

$$\frac{(\beta_2 - \beta_1)(1 - \beta_1)}{1 - \beta_2} \sum_{\tau=t_0}^{t-1} \left( \eta_\tau \frac{1 - \beta_1^{\tau - t_0}}{1 - \beta_1} - \sum_{\tau'=1}^{t-\tau-1} \eta_{\tau+\tau'} \beta_1^{\tau'-1} \right) \log(\mathbf{v}_\tau[k])$$

$$\leq \frac{\beta_2 - \beta_1}{1 - \beta_2} \sum_{\tau=t_0}^{t-1} \eta_\tau \beta_1^{\tau-t_0} \big( -\tau \log \beta_2 - \log(1 - \beta_2) - \log \rho \big)$$

$$\leq \frac{\eta_{t_0}(\beta_2 - \beta_1)}{(1 - \beta_1)(1 - \beta_2)} \left[ \Big( t_0 + \frac{1}{1 - \beta_1} \Big)(-\log \beta_2) - \log(1 - \beta_2) - \log \rho \right].$$

Plugging these results into (A.5) with Bernoulli inequality, we have,

$$|\mathbf{w}_t[k]| \leq |\mathbf{w}_{t_0}[k]| + \sum_{\tau=t_0}^{t-1} \eta_\tau + \frac{\eta_{t_0}(\beta_2 - \beta_1)}{2(1 - \beta_1)(1 - \beta_2)} \left[ \Big( t_0 + \frac{1}{1 - \beta_1} \Big)(-\log \beta_2) - \log(1 - \beta_2) - \log \rho + 1 \right]$$

$$\leq \sum_{\tau=t_0}^{t-1} \eta_\tau + \alpha \sum_{\tau=0}^{t_0-1} \eta_\tau + \frac{\eta_{t_0}(\beta_2 - \beta_1)}{2(1 - \beta_1)(1 - \beta_2)} \left[ \Big( t_0 + \frac{1}{1 - \beta_1} \Big)(-\log \beta_2) - \log(1 - \beta_2) - \log \rho + 1 \right]$$

$$\leq \sum_{\tau=t_0}^{t-1} \eta_\tau + \alpha \sum_{\tau=0}^{t_0-1} \eta_\tau + C_6' \eta_0 t_0 \leq \sum_{\tau=t_0}^{t-1} \eta_\tau + C_6 \sum_{\tau=0}^{t_0-1} \eta_\tau,$$

where $C_6$ and $C_6'$ are constants only depending on $\beta_1, \beta_2$ and $B$. The second inequality is from triangle inequality of $|\mathbf{w}_{t_0}[k]|$ and Lemma 6.5. The third inequality is from our condition $t_0 > -\log \rho$ and the last inequality is because $\eta_t$ is decreasing. Since the preceding result holds for all $k \in [d]$, it also holds for $\|\mathbf{w}_t\|_\infty$, which finishes the proof. $\qquad\square$

## B  Complete Proof for Theorem 4.5 and Calculation Details for Corollary 4.7

### B.1  Complete Proof for Theorem 4.5

*Proof of Theorem 4.5.* For $C_1, C_2$ defined in Lemma 6.2, it's trivial that when $t$ is large we have the following inequalities hold:(i).$\beta_1^{t/2} \leq \frac{1}{6C_1}$; (ii).$\eta_t \leq \min\big\{ \frac{\gamma^2}{36C_2^2 d^2}, \frac{\gamma}{6C_2 d} \big\}$. We use $t_2 = t_2(d, \beta_1, \beta_2, \gamma, B)$ to denote the first time that all the preceding inequalities hold after $t_1 = t_1(\beta_1, \beta_2, B)$ in Assumption 4.4. Plugging all aforementioned inequality conditions into Lemma 6.2, we can derive that for all $t \geq t_2$,

$$\mathcal{R}(\mathbf{w}_{t+1}) \leq \mathcal{R}(\mathbf{w}_t) - \frac{\eta_t \gamma}{2} \mathcal{G}(\mathbf{w}_t). \tag{B.1}$$

Therefore we prove that $\mathcal{R}(\mathbf{w}_{t+1}) < \mathcal{R}(\mathbf{w}_t)$ for all $t \geq t_2$. For further proof, we separately consider $\ell = \ell_{\exp}$ and $\ell = \ell_{\log}$.

When $\ell = \ell_{\exp}$, by definition we have $\mathcal{G}(\mathbf{w}_t) = \mathcal{R}(\mathbf{w}_t)$. Therefore, for all $t \geq t_2$, (B.1) can be re-written as

$$\mathcal{R}(\mathbf{w}_{t+1}) \leq \Big( 1 - \frac{\gamma \eta_t}{2} \Big) \cdot \mathcal{R}(\mathbf{w}_t) \leq \mathcal{R}(\mathbf{w}_t) \cdot e^{-\frac{\gamma \eta_t}{2}} \leq \mathcal{R}(\mathbf{w}_{t_2}) \cdot e^{-\frac{\gamma \sum_{\tau=t_2}^{t} \eta_\tau}{2}}.$$

Although $\ell_{\exp}$ is not Lipschitz continuous over $\mathbb{R}$, we have

$$\mathcal{R}(\mathbf{w}_{t_2}) \leq \mathcal{R}(\mathbf{w}_0) \cdot \exp \Big( \frac{1}{n} \sum_{i=1}^{n} \|\mathbf{x}_i\|_1 \|\mathbf{w}_{t_2} - \mathbf{w}_0\|_\infty \Big) \leq \mathcal{R}(\mathbf{w}_0) \cdot \exp \Big( \alpha B \sum_{\tau=0}^{t_2-1} \eta_\tau \Big)$$

by Lemma 6.5 and triangle inequality. Letting $\mathcal{R}_0 = \min\{ \frac{\log 2}{n}, \frac{1}{\sqrt{B^2 + c_v^2 \eta_0}} \}$ and $t_0 = t_0(n, d, \beta_1, \beta_2, \gamma, B, t_1, \mathbf{w}_0)$ be the first time be the first time such that $\sum_{\tau=t_2}^{t_0-1} \eta_\tau \geq \frac{2\alpha B}{\gamma} \sum_{\tau=0}^{t_2-1} \eta_\tau + \frac{2 \log \mathcal{R}(\mathbf{w}_0) - 2 \log \mathcal{R}_0}{\gamma}$ and $t_0 \geq -\log \rho$. By such definition of $t_0$, we can derive that for all $t \geq t_0$,

$$\mathcal{R}(\mathbf{w}_t) \leq \mathcal{R}(\mathbf{w}_{t_2}) \cdot e^{-\frac{\gamma \sum_{\tau=t_0}^{t} \eta_\tau}{2}} \cdot e^{-\frac{\gamma \sum_{\tau=t_2}^{t_0-1} \eta_\tau}{2}} \leq \mathcal{R}_0 \cdot e^{-\frac{\gamma \sum_{\tau=t_0}^{t} \eta_\tau}{2}}.$$

Since $t_0$ satisfies all the requirements in Lemma 6.3 and Lemma 6.4, we can finally derive that

$$\left| \min_{i \in [n]} \frac{\langle \mathbf{w}_t, y_i \cdot \mathbf{x}_i \rangle}{\|\mathbf{w}_t\|_\infty} - \gamma \right| \leq \frac{\gamma C_6 \sum_{\tau=0}^{t_0-1} \eta_\tau + C_3 d \Big( \sum_{\tau=t_0}^{t-1} \eta_\tau^{\frac{3}{2}} + \sum_{\tau=t_0}^{t-1} \eta_\tau^2 \Big) + C_4}{\sum_{\tau=t_0}^{t-1} \eta_\tau + C_6 \sum_{\tau=0}^{t_0-1} \eta_\tau}$$

$$\leq O\left(\frac{\sum_{\tau=0}^{t_0-1}\eta_\tau + d\sum_{\tau=t_0}^{t-1}\eta_\tau^{\frac{3}{2}}}{\sum_{\tau=0}^{t-1}\eta_\tau}\right),$$

since the decay learning rate $\eta_t \to 0$ by Assumption 4.3, and $C_3, C_4$ and $C_6$ are constants solely depending on $\beta_1, \beta_2$ and $B$.

When $\ell = \ell_{\log}$, by taking a telescoping sum on the result of (B.1), we obtain $\gamma \sum_{\tau=t_2}^{t}\eta_\tau \mathcal{G}(\mathbf{w}_t) \leq 2\mathcal{R}(\mathbf{w}_{t_2})$ for any $t \geq t_2$. Since $\ell_{\log}$ is 1-Lipschitz continuous, we can derive that $\mathcal{R}(\mathbf{w}_{t_1}) \leq \mathcal{R}(\mathbf{w}_0) + \alpha B \sum_{\tau=0}^{t_2-1}\eta_\tau$. Letting $\mathcal{R}_0 = \min\{\frac{\log 2}{n}, \frac{1}{\sqrt{B^2+c_v^2\eta_0}}\}$ and $t_0 = t_0(n, d, \beta_1, \beta_2, \gamma, B, t_1, \mathbf{w}_0)$ be the first time such that $\sum_{\tau=t_2}^{t_0-1}\eta_\tau \geq \frac{4\mathcal{R}(\mathbf{w}_0)+4\alpha B\sum_{\tau=0}^{t_2-1}\eta_\tau}{\gamma\mathcal{R}_0}$ and $t_0 \geq -\log\rho$, then we can conclude that

$$\min_{\tau\in[t_2,t_0]}\mathcal{G}(\mathbf{w}_\tau) \leq \frac{2\mathcal{R}(\mathbf{w}_{t_2})}{\gamma\sum_{\tau=t_2}^{t_0-1}\eta_\tau} \leq \frac{2\mathcal{R}(\mathbf{w}_0)+2\alpha B\sum_{\tau=0}^{t_2-1}\eta_\tau}{\gamma\sum_{\tau=t_2}^{t_0-1}\eta_\tau} \leq \frac{\mathcal{R}_0}{2}.$$

Let $\tau' = \operatorname{argmin}_{\tau\in[t_2,t_0]}\mathcal{G}(\mathbf{w}_\tau)$, then we obtain that $\langle\mathbf{z}_i, \mathbf{w}_{\tau'}\rangle \geq 0$ for all $i \in [n]$ by Lemma C.4. Moreover, by Lemma C.3 and the monotonicity of $\mathcal{R}(\mathbf{w}_t)$ derived in (B.1), we can conclude that $\mathcal{R}(\mathbf{w}_t) < \mathcal{R}(\mathbf{w}_{\tau'}) \leq 2\mathcal{G}(\mathbf{w}_{\tau'}) \leq \mathcal{R}_0$ for all $t > t_0$. Similarly, the inequality $\mathcal{R}(\mathbf{w}_t) < \mathcal{R}_0$ also implies $\langle\mathbf{z}_i, \mathbf{w}_t\rangle \geq 0$ for all $t > t_0$ by Lemma C.4, and correspondingly $\mathcal{R}(\mathbf{w}_t) \leq 2\mathcal{G}(\mathbf{w}_t)$ by Lemma C.3. Then for all $t \geq t_0$, we can re-write (B.1) as

$$\mathcal{R}(\mathbf{w}_t) \leq \mathcal{R}(\mathbf{w}_{t-1}) - \frac{\gamma\eta_{t-1}}{2}\cdot\mathcal{G}(\mathbf{w}_t) \leq \left(1 - \frac{\gamma\eta_{t-1}}{4}\right)\cdot\mathcal{R}(\mathbf{w}_{t-1})$$

$$\leq \mathcal{R}(\mathbf{w}_{t-1})\cdot e^{-\frac{\gamma\eta_{t-1}}{4}} \leq \mathcal{R}(\mathbf{w}_{t_0})\cdot e^{-\frac{\gamma}{4}\sum_{\tau=t_0}^{t-1}\eta_\tau} \leq \mathcal{R}_0\cdot e^{-\frac{\gamma}{4}\sum_{\tau=t_0}^{t-1}\eta_\tau}.$$

By this result and Lemma C.7, we have

$$\frac{\mathcal{G}(\mathbf{w}_t)}{\mathcal{R}(\mathbf{w}_t)} \geq 1 - \frac{n\mathcal{R}(\mathbf{w}_t)}{2} \geq 1 - \frac{n\mathcal{R}_0\cdot e^{-\frac{\gamma}{4}\sum_{\tau=t_0}^{t-1}\eta_\tau}}{2} \geq 1 - e^{-\frac{\gamma}{4}\sum_{\tau=t_0}^{t-1}\eta_\tau}. \tag{B.2}$$

Since $t_0$ satisfies all the requirements in Lemma 6.3 and Lemma 6.4, we can combine Lemma 6.3, Lemma 6.4 and (B.2) and finally derive that

$$\left|\min_{i\in[n]}\frac{\langle\mathbf{w}_t, y_i\cdot\mathbf{x}_i\rangle}{\|\mathbf{w}_t\|_\infty} - \gamma\right| \leq \frac{\gamma C_6\sum_{\tau=0}^{t_0-1}\eta_\tau + C_3 d\left(\sum_{\tau=t_0}^{t-1}\eta_\tau^{\frac{3}{2}} + \sum_{\tau=t_0}^{t-1}\eta_\tau^2\right) + C_4 + \sum_{\tau=t_0}^{t-1}\eta_\tau e^{-\frac{\gamma}{4}\sum_{\tau'=t_0}^{\tau-1}\eta_{\tau'}'}}{\sum_{\tau=t_0}^{t-1}\eta_\tau + C_6\sum_{\tau=0}^{t_0-1}\eta_\tau}$$

$$\leq O\left(\frac{\sum_{\tau=0}^{t_0-1}\eta_\tau + d\sum_{\tau=t_0}^{t-1}\eta_\tau^{\frac{3}{2}} + \sum_{\tau=t_0}^{t-1}\eta_\tau e^{-\frac{\gamma}{4}\sum_{\tau'=t_0}^{\tau-1}\eta_{\tau'}'}}{\sum_{\tau=0}^{t-1}\eta_\tau}\right),$$

since the decay learning rate $\eta_t \to 0$ by Assumption 4.3, and $C_3, C_4$ and $C_6$ are constants solely depending on $\beta_1, \beta_2$ and $B$. $\qquad\square$

### B.2 Calculation Details for Corollary 4.7

In this section, we use the notation $C_1, C_2, C_3, \ldots$ to denote constants solely depending on $\beta_1, \beta_2, \gamma$ and $B$. While it may seem an abuse of notation as these symbols could be different from Section 6 or denote distinct constants across different formulas, we assert that their exact values are immaterial for our analysis. Therefore, we opt for this shorthand notation for the sake of brevity and clarity, without concern for the precise numerical values of these constants in each instance.

For given $\eta_t = (t+2)^{-a}$ with $a \in (0, 1]$, recall the definition of $t_2$ in Appendix B.1 to be the first time such that (i).$\beta_1^{t/2} \leq \frac{1}{6C_1}$; (ii).$\eta_t \leq \min\left\{\frac{\gamma^2}{36C_2^2 d^2}, \frac{\gamma}{6C_2 d}\right\}$. We can derive that

$$t_2 = \max\left\{-\frac{2\log 6 + 2\log C_1}{\log\beta_1}, \left(\frac{36C_2^2 d^2}{\gamma^2}\right)^{\frac{1}{a}}, \left(\frac{6C_2 d}{\gamma}\right)^{\frac{1}{a}}\right\} = C_3 d^{\frac{2}{a}}.$$

We consider the following four cases,

- If $\ell = \ell_{\exp}$ and $a \in (0,1)$, recall in Appendix B.1, the definition for $t_0$ when $\ell = \ell_{\exp}$ is the first time such that $\sum_{\tau=t_2}^{t_0-1} \eta_\tau \geq \frac{2\alpha B}{\gamma} \sum_{\tau=0}^{t_2-1} \eta_\tau + \frac{2\log \mathcal{R}(\mathbf{w}_0) - 2\log \mathcal{R}_0}{\gamma}$ and $t_0 \geq -\log \rho$, by simple approximation from integral of $t^{-a}$, we can derive that,

$$t_0 \leq C_1 d^{\frac{2}{a}} + C_2 [\log n]^{\frac{1}{1-a}} + C_3 [\log \mathcal{R}(\mathbf{w}_0)]^{\frac{1}{1-a}} + C_4 \log(1/\rho).$$

Similarly, we can also derive

$$\sum_{\tau=0}^{t_0-1} \eta_\tau \leq C_1 t_0^{1-a} \leq C_2 d^{\frac{2(1-a)}{a}} + C_3 \log n + C_4 \log \mathcal{R}(\mathbf{w}_0) + C_5 [\log(1/\rho)]^{1-a}.$$

When $a > \frac{2}{3}$, $\sum_{\tau=t_0}^{\infty} \eta_\tau^{\frac{3}{2}}$ is bounded by some constant, $\sum_{\tau=0}^{t-1} \eta_\tau = O(t^{1-a})$ and $\frac{2(1-a)}{a} < 1$, therefore we conclude that,

$$\left| \min_{i \in [n]} \frac{\langle \mathbf{w}_t^{\exp}, \mathbf{z}_i \rangle}{\|\mathbf{w}_t^{\exp}\|_\infty} - \gamma \right| \leq O\left( \frac{d + \log n + \log \mathcal{R}(\mathbf{w}_0) + [\log(1/\rho)]^{1-a}}{t^{1-a}} \right).$$

When $a = \frac{2}{3}$, similarly we have $\sum_{\tau=t_0}^{t-1} \eta_\tau^{\frac{3}{2}} = O(\log t)$ and $\sum_{\tau=0}^{t-1} \eta_\tau = O(t^{1/3})$, then we conclude that

$$\left| \min_{i \in [n]} \frac{\langle \mathbf{w}_t^{\exp}, \mathbf{z}_i \rangle}{\|\mathbf{w}_t^{\exp}\|_\infty} - \gamma \right| \leq O\left( \frac{d \cdot \log t + \log n + \log \mathcal{R}(\mathbf{w}_0) + [\log(1/\rho)]^{1/3}}{t^{1/3}} \right).$$

When $a < \frac{2}{3}$, similarly we have $\sum_{\tau=t_0}^{t-1} \eta_\tau^{\frac{3}{2}} = O(t^{1-\frac{3a}{2}})$ and $\sum_{\tau=0}^{t-1} \eta_\tau = O(t^{1-a})$. Under this sub-case, we could always find a new $t_0' \geq t_0$ such that besides the preceding condition for $t_0$, we also have $d \cdot t_0'^{1-3a/2} > \max\{d^{\frac{2(1-a)}{a}}, \log n, \log \mathcal{R}(\mathbf{w}_0), [\log(1-\rho)]^{1-a}\}$. Letting this new $t_0'$ to be the $t_0$ in our statement of Corollary 4.7, then we conclude that,

$$\left| \min_{i \in [n]} \frac{\langle \mathbf{w}_t^{\exp}, \mathbf{z}_i \rangle}{\|\mathbf{w}_t^{\exp}\|_\infty} - \gamma \right| \leq O\left( \frac{d \cdot t^{1-\frac{3a}{2}} + d^{\frac{2(1-a)}{a}} + \log n + \log \mathcal{R}(\mathbf{w}_0) + [\log(1/\rho)]^{1-a}}{t^{1-a}} \right) \leq O\left( \frac{d}{t^{a/2}} \right).$$

- If $\ell = \ell_{\exp}$ and $a = 1$, then by the definition of $t_0$ and integral of $t^{-1}$, we obtain that,

$$\log t_0 \leq C_1 \log d + C_2 \log n + C_3 \log \mathcal{R}(\mathbf{w}_0) + C_4 \log \log(1/\rho).$$

Similarly, we can also derive

$$\sum_{\tau=0}^{t_0-1} \eta_\tau \leq C_1 \log d + C_2 \log n + C_3 \log \mathcal{R}(\mathbf{w}_0) + C_4 \log \log(1/\rho).$$

Since $\sum_{\tau=0}^{t-1} \eta_\tau = O(\log t)$ and $\sum_{\tau=t_0}^{t-1} \eta_\tau^{\frac{3}{2}}$ is bounded, we obtain that,

$$\left| \min_{i \in [n]} \frac{\langle \mathbf{w}_t^{\exp}, \mathbf{z}_i \rangle}{\|\mathbf{w}_t^{\exp}\|_\infty} - \gamma \right| \leq O\left( \frac{d + \log n + \log \mathcal{R}(\mathbf{w}_0) + \log \log(1/\rho)}{\log t} \right).$$

- If $\ell = \ell_{\log}$ and $a \in (0,1)$, firstly we upper bound the last term $\sum_{\tau=t_0}^{t} \eta_\tau e^{-\frac{\gamma}{4} \sum_{\tau'=t_0}^{\tau-1} \eta_{\tau'}}$ as

$$\sum_{\tau=t_0}^{t-1} \eta_\tau e^{-\frac{\gamma}{4} \sum_{\tau'=t_0}^{\tau-1} \eta_{\tau'}} \leq e^{\frac{\gamma(t_0+1)^{1-a}}{4(1-a)}} \sum_{\tau=t_0}^{\infty} \frac{1}{(\tau+2)^a} e^{-\frac{\gamma(\tau+1)^{1-a}}{4(1-a)}} \leq \frac{4}{\gamma} e^{\frac{\gamma}{4(1-a)}}.$$

Therefore, $\sum_{\tau=t_0}^{t} \eta_\tau e^{-\frac{\gamma}{4} \sum_{\tau'=t_0}^{\tau-1} \eta_{\tau'}}$ is always bounded by a constant. The only difference between $\ell_{\log}$ and $\ell_{\exp}$ is how to determine the value of $t_0$. For $\ell_{\log}$, the formula for $t_0$ is the first time such that $\sum_{\tau=t_2}^{t_0-1} \eta_\tau \geq \frac{4\mathcal{R}(\mathbf{w}_0) + 4\alpha B \sum_{\tau=0}^{t_2-1} \eta_\tau}{\gamma \mathcal{R}_0}$ and $t \geq \log(1/\rho)$. Similar to the preceding process, we could derive that

$$t_0 \leq C_1 n^{\frac{1}{1-a}} d^{\frac{2}{a}} + C_2 n^{\frac{1}{1-a}} [\mathcal{R}(\mathbf{w}_0)]^{\frac{1}{1-a}} + C_3 \log(1/\rho).$$

and

$$\sum_{\tau=0}^{t_0-1} \eta_\tau \le C_1 n d^{\frac{2(1-a)}{a}} + C_2 n \mathcal{R}(\mathbf{w}_0) + C_3 [\log(1/\rho)]^{1-a}.$$

When $a > \frac{2}{3}$, $\sum_{\tau=t_0}^{\infty} \eta_\tau^{\frac{3}{2}}$ is bounded by some constant, $\sum_{\tau=0}^{t-1} \eta_\tau = O(t^{1-a})$ and $\frac{2(1-a)}{a} \le 1$, therefore we conclude that

$$\left| \min_{i\in[n]} \frac{\langle \mathbf{w}_t^{\log}, \mathbf{z}_i \rangle}{\|\mathbf{w}_t^{\log}\|_\infty} - \gamma \right| \le O\left( \frac{d + nd^{\frac{2(1-a)}{a}} + n\mathcal{R}(\mathbf{w}_0) + [\log(1/\rho)]^{1-a}}{t^{1-a}} \right).$$

When $a = \frac{2}{3}$, similarly we have $\sum_{\tau=t_0}^{t-1} \eta_\tau^{\frac{3}{2}} = O(\log t)$ and $\sum_{\tau=0}^{t-1} \eta_\tau = O(t^{1/3})$, then we conclude that

$$\left| \min_{i\in[n]} \frac{\langle \mathbf{w}_t^{\log}, \mathbf{z}_i \rangle}{\|\mathbf{w}_t^{\log}\|_\infty} - \gamma \right| \le O\left( \frac{d \cdot \log t + nd + n\mathcal{R}(\mathbf{w}_0) + [\log(1/\rho)]^{1/3}}{t^{1/3}} \right).$$

When $a < \frac{2}{3}$, similarly we have $\sum_{\tau=t_0}^{t-1} \eta_\tau^{\frac{3}{2}} = O(t^{1-\frac{3a}{2}})$ and $\sum_{\tau=0}^{t-1} \eta_\tau = O(t^{1-a})$. Under this setting, we could always find a new $t_0' \ge t_0$ such that besides the preceding condition for $t_0$, we also have $d \cdot t_0'^{1-3a/2} > \max\{nd^{\frac{2(1-a)}{a}}, n\log\mathcal{R}(\mathbf{w}_0), [\log(1-\rho)]^{1-a}\}$. Letting this new $t_0'$ to be the $t_0$ in our statement of Corollary 4.7, then we conclude that,

$$\left| \min_{i\in[n]} \frac{\langle \mathbf{w}_t^{\log}, \mathbf{z}_i \rangle}{\|\mathbf{w}_t^{\log}\|_\infty} - \gamma \right| \le O\left( \frac{d \cdot t^{1-\frac{3a}{2}} + nd^{\frac{2(1-a)}{a}} + n\mathcal{R}(\mathbf{w}_0) + [\log(1/\rho)]^{1-a}}{t^{1-a}} \right) \le O\left( \frac{d}{t^{a/2}} \right).$$

- If $\ell = \ell_{\log}$ and $a = 1$, firstly we upper bound the last term $\sum_{\tau=t_0}^{t} \eta_\tau e^{-\frac{\gamma}{4}\sum_{\tau'=t_0}^{\tau-1} \eta_{\tau'}}$ as

$$\sum_{\tau=t_0}^{t-1} \eta_\tau e^{-\frac{\gamma}{4}\sum_{\tau'=t_0}^{\tau-1} \eta_{\tau'}} \le (t_0+1)^{\gamma/4} \sum_{\tau=t_0}^{\infty} \frac{1}{(\tau+1)^{1+\gamma/4}} \le \frac{\gamma}{4} 2^{\gamma/4},$$

which implies $\sum_{\tau=t_0}^{t} \eta_\tau e^{-\frac{\gamma}{4}\sum_{\tau'=t_0}^{\tau-1} \eta_{\tau'}}$ is also a constant. Then by the definition of $t_0$ and integral of $t^{-1}$, we obtain that

$$\log t_0 \le C_1 n \log d + C_2 n\mathcal{R}(\mathbf{w}_0) + C_3 \log\log(1/\rho).$$

and similarly

$$\sum_{\tau=0}^{t_0-1} \le C_1 n \log d + C_2 n\mathcal{R}(\mathbf{w}_0) + C_3 \log\log(1/\rho).$$

Since $\sum_{\tau=0}^{t-1} \eta_\tau = O(\log t)$ and $\sum_{\tau=t_0}^{t-1} \eta_\tau^{\frac{3}{2}}$ is bounded, we obtain that

$$\left| \min_{i\in[n]} \frac{\langle \mathbf{w}_t^{\log}, \mathbf{z}_i \rangle}{\|\mathbf{w}_t^{\log}\|_\infty} - \gamma \right| \le O\left( \frac{d + n\log d + n\mathcal{R}(\mathbf{w}_0) + \log\log(1/\rho)}{\log t} \right).$$

## C   Technical Lemmas

### C.1   Lemma for Assumption 4.4

**Lemma C.1.** Assumption 4.4 holds for both small fixed learning rate $\eta_t = \eta \le \frac{1-\beta}{2c_1}$, and decay learning rate $\eta_t = (t+2)^{-a}$ with $a \in (0,1]$.

*Proof of Lemma C.1.* Firstly, we prove it for learning rate $\eta_t = \eta \le \frac{1-\beta}{2c_1}$, which is a small fixed constant, then we have,

$$\sum_{\tau=0}^{t} \beta^\tau \left( e^{c_1 \sum_{\tau'=1}^{\tau} \eta_{t-\tau'}} - 1 \right) = \sum_{\tau=0}^{t} \beta^\tau \left( e^{c_1 \eta \tau} - 1 \right) = \sum_{\tau=0}^{t} \beta^\tau \sum_{k=1}^{\infty} \frac{(c_1 \eta \tau)^k}{k!}$$

$$\leq \sum_{\tau=0}^{\infty} \beta^\tau \sum_{k=1}^{\infty} \frac{(c_1\eta\tau)^k}{k!} = \sum_{k=1}^{\infty} \frac{(c_1\eta)^k}{k!} \sum_{\tau=0}^{\infty} \beta^\tau \tau^k$$

$$\leq \frac{1}{1-\beta} \sum_{k=1}^{\infty} \Big(\frac{c_1\eta}{1-\beta}\Big)^k \leq \frac{2c_1}{(1-\beta)^2}\eta$$

The penultimate inequality holds because $\sum_{\tau=0}^{\infty} \beta^\tau \tau^k \leq \int_0^\infty \beta^\tau \tau^k \mathrm{d}\tau = \frac{k!}{[-\log(\beta)]^{k+1}} \leq \frac{k!}{(1-\beta)^{k+1}}$. Therefore, we prove that Assumption 4.4 holds for $t \in \mathbb{N}_+$ when $\eta_t = \eta \leq \frac{1-\beta}{2c_1}$. Next, we consider the case where decay learning rate $\eta_t = \frac{1}{(t+2)^a}$ with $a \in (0,1)$, and similarly, we have,

$$\sum_{\tau'=1}^{\tau} \eta_{t-\tau'} = \sum_{\tau'=1}^{\tau} \frac{1}{(t-\tau'+2)^a} \leq \int_1^{\tau+1} \frac{1}{(t-\tau'+2)^a}\mathrm{d}\tau'$$

$$= \frac{1}{1-a}\Big((t+1)^{1-a} - (t-\tau+1)^{1-a}\Big)$$

$$= \frac{1}{1-a}\frac{\tau + (t+1)^{1-a}(t-\tau+1)^a - (t+1)^a(t-\tau+1)^{1-a}}{(t+1)^a + (t-\tau+1)^a}$$

$$\leq \frac{2}{(1-a)}\frac{\tau}{(t+1)^a}$$

By this result, we can similarly obtain that for $t \geq \Big(\frac{4c_1}{(1-\beta)^2(1-a)}\Big)^{\frac{1}{a}}$.

$$\sum_{\tau=0}^{t} \beta^\tau \big(e^{c_1\sum_{\tau'=1}^{\tau}\eta_{t-\tau'}} - 1\big) \leq \sum_{\tau=0}^{t} \beta^\tau \big(e^{\frac{2c_1\tau}{(1-a)(t+1)^a}} - 1\big) = \sum_{\tau=0}^{t} \beta^\tau \sum_{k=1}^{\infty} \Big(\frac{2c_1\tau}{(1-a)(t+1)^a}\Big)^k \frac{1}{k!}$$

$$\leq \sum_{\tau=0}^{\infty} \beta^\tau \sum_{k=1}^{\infty} \Big(\frac{2c_1\tau}{(1-a)(t+1)^a}\Big)^k \frac{1}{k!} = \sum_{k=1}^{\infty} \Big(\frac{2c_1}{(1-a)(t+1)^a}\Big)^k \frac{1}{k!} \sum_{\tau=0}^{\infty} \beta^\tau \tau^k$$

$$\leq \frac{1}{1-\beta} \sum_{k=1}^{\infty} \Big(\frac{2c_1}{(1-\beta)(1-a)(t+1)^a}\Big)^k$$

$$\leq \frac{4c_1}{(1-\beta)^2(1-a)} \cdot \frac{1}{(t+1)^a} < \frac{8c_1}{(1-\beta)^2(1-a)} \cdot \eta_t.$$

Therefore, we prove that Assumption 4.4 holds for $t \geq \Big(\frac{4c_1}{(1-\beta)^2(1-a)}\Big)^{\frac{1}{a}}$ when $\eta_t = \frac{1}{(t+2)^a}$. Finally we consider the case where the decay learning rate $\eta_t = \frac{1}{t+2}$, and similarly, we have $\sum_{\tau'=1}^{\tau} \eta_{t-\tau'} = \sum_{\tau'=1}^{\tau} \frac{1}{t-\tau'+2} \leq \int_1^{\tau+1} \frac{1}{t-\tau'+2}\mathrm{d}\tau' = \log(1 + \frac{\tau}{t-\tau+1})$. By this result, we obtain that

$$\sum_{\tau=0}^{t} \beta^\tau \big(e^{c_1\sum_{\tau'=1}^{\tau}\eta_{t-\tau'}} - 1\big) \leq \sum_{\tau=0}^{t} \beta^\tau \Big(\Big(1 + \frac{\tau}{t-\tau+1}\Big)^{\lceil c_1\rceil} - 1\Big)$$

$$= \sum_{\tau=0}^{t} \beta^\tau \sum_{k=1}^{\lceil c_1\rceil} \binom{\lceil c_1\rceil}{k}\Big(\frac{\tau}{t-\tau+1}\Big)^k. \tag{C.1}$$

Because $\lceil c_1\rceil$ and $\binom{\lceil c_1\rceil}{k}$ are both absolute constant, it suffices to show that $\sum_{\tau=0}^{t} \beta^\tau \Big(\frac{\tau}{t-\tau+1}\Big)^k \leq \frac{c_2}{t+2}$ for all $t > t_1$ and $k \geq 1$ where $t_1, c_2$ are both constants. Actually, we could split the summation of $\tau$ into two parts as

$$\sum_{\tau=0}^{t} \beta^\tau \Big(\frac{\tau}{t-\tau+1}\Big)^k = \sum_{\tau=0}^{\lfloor\frac{t}{2}\rfloor} \beta^\tau \Big(\frac{\tau}{t-\tau+1}\Big)^k + \sum_{\tau=\lfloor\frac{t}{2}\rfloor+1}^{t} \beta^\tau \Big(\frac{\tau}{t-\tau+1}\Big)^k.$$

For the first part, we have

$$\sum_{\tau=0}^{\lfloor \frac{t}{2} \rfloor} \beta^\tau \left(\frac{\tau}{t-\tau+1}\right)^k \leq \left(\frac{2}{t}\right)^k \sum_{\tau=0}^{\lfloor \frac{t}{2} \rfloor} \beta^\tau \tau^k \leq \frac{2^k k!}{(1-\beta)^{k+1}} \cdot \frac{1}{t}.$$

For the second part, we could find a constant $t_1 = t_1(c_1, \beta)$ such that $t^{\lceil c_1 \rceil + 2} \leq \left(\frac{1}{\beta}\right)^{\frac{t}{2}}$ for all $t \geq t_1$ since $\frac{1}{\beta} > 1$. Then for all $t \geq t_1$, we can derive that

$$\sum_{\tau=\lfloor \frac{t}{2} \rfloor + 1}^{t} \beta^\tau \left(\frac{\tau}{t-\tau+1}\right)^k \leq \beta^{\frac{t}{2}} \sum_{\tau=\lfloor \frac{t}{2} \rfloor + 1}^{t} \tau^k \leq \beta^{\frac{t}{2}} t^{k+1} \leq \frac{1}{t}.$$

The last inequality is by our condition $t \geq t_1$ and $k \leq \lceil c_1 \rceil$. Combining these two results and plugging it into (C.1), we finally get

$$\sum_{\tau=0}^{t} \beta^\tau \left(e^{c_1 \sum_{\tau'=1}^{\tau} \eta_{t-\tau'}} - 1\right) \leq \lceil c_1 \rceil \cdot \max_{k \in [\lceil c_1 \rceil]} \binom{\lceil c_1 \rceil}{k} \cdot \left(\frac{2^{\lceil c_1 \rceil} \lceil c_1 \rceil!}{(1-\beta)^{\lceil c_1 \rceil + 1}} + 1\right) \cdot \frac{1}{t}$$

$$\leq 2\lceil c_1 \rceil \cdot \max_{k \in [\lceil c_1 \rceil]} \binom{\lceil c_1 \rceil}{k} \cdot \left(\frac{2^{\lceil c_1 \rceil} \lceil c_1 \rceil!}{(1-\beta)^{\lceil c_1 \rceil + 1}} + 1\right) \cdot \eta_t$$

for all $t \geq t_1$, which completes the proof. $\qquad\square$

## C.2 Properties for Logistic and Exponential Loss Function

**Lemma C.2.** For $\ell \in \{\ell_{\exp}, \ell_{\log}\}$ and any $z \in \mathbb{R}$, $\ell''(z) \leq |\ell'(z)| \leq \ell(z)$. For any $z \geq 0$, $\ell_{\log}(z) \leq 2|\ell'_{\log}(z)|$.

*Proof of Lemma C.2.* For $\ell = \ell_{\exp}$, $|\ell'_{\exp}(z)| = \ell''_{\exp}(z) = \ell_{\exp}(z) = e^{-z}$. For $\ell = \ell_{\log}$, we calculate the derivatives as $\ell'_{\log}(z) = -\frac{1}{1+e^z}$ and $\ell''_{\log}(z) = \frac{e^z}{(1+e^z)^2}$. Notice that

$$\lim_{z \to +\infty} \ell_{\log}(z) = \lim_{z \to +\infty} |\ell'_{\log}(z)| = 0,$$

and

$$\ell'_{\log}(z) = -\frac{1}{1+e^z} \leq -\frac{1}{2+e^z+e^{-z}} = -\ell''_{\log}(z) = \left(|\ell'_{\log}(z)|\right)'.$$

Therefore we derive that $\ell''_{\log}(z) \leq |\ell'_{\log}(z)| \leq \ell_{\log}(z)$. $\qquad\square$

**Lemma C.3.** For any $z \geq 0$, $\frac{|\ell'_{\log}(z)|}{\ell_{\log}(z)}, \frac{|\ell'_{\log}(z)|}{\ell_{\exp}(z)} \geq \frac{1}{2}$ and $\frac{\ell_{\log}(z)}{\ell_{\exp}(z)} \geq \log 2$.

*Proof of Lemma C.3.* For $z \geq 0$, it holds that

$$\ell_{\log}(z) \leq \ell_{\exp}(z) = \frac{2}{2e^z} \leq \frac{2}{1+e^z} = 2|\ell'_{\log}(z)|.$$

The second result holds because $\frac{\ell_{\log}(z)}{\ell_{\exp}(z)} = \frac{\log(1+e^{-z})}{e^{-z}}$ is an decreasing function for $e^{-z}$ and $e^{-z} \in (0,1]$ for $z \geq 0$. $\qquad\square$

**Lemma C.4.** For $\ell \in \{\ell_{\exp}, \ell_{\log}\}$, either $\mathcal{G}(\mathbf{w}) \leq \frac{1}{2n}$ or $\mathcal{R}(\mathbf{w}) \leq \frac{\log 2}{n}$ implies $\langle \mathbf{w}, \mathbf{z}_i \rangle \geq 0$ for all $i \in [n]$.

*Proof of Lemma C.4.* If $\mathcal{G}(\mathbf{w}) \leq \frac{1}{2n}$, we have $\left|\ell'(\langle \mathbf{w}, \mathbf{z}_i \rangle)\right| \leq n\mathcal{G}(\mathbf{w}) \leq \frac{1}{2}$. Then by monotonicity of $|\ell'(\cdot)|$ we have $\langle \mathbf{w}, \mathbf{z}_i \rangle \geq 0$. Similarly if $\mathcal{R}(\mathbf{w}) \leq \frac{\log 2}{n}$, we also have $\ell(\langle \mathbf{w}, \mathbf{z}_i \rangle) \leq n\mathcal{R}(\mathbf{w}) \leq \log 2$. Then by monotonicity of $\ell(\cdot)$ we have $\langle \mathbf{w}, \mathbf{z}_i \rangle \geq 0$. $\qquad\square$

**Lemma C.5.** For $\ell \in \{\ell_{\exp}, \ell_{\log}\}$ and any $z_1, z_2 \in \mathbb{R}$, we have

$$\left|\frac{\ell'(z_1)}{\ell'(z_2)} - 1\right| \leq e^{|z_1 - z_2|} - 1 \tag{C.2}$$

*Proof of Lemma C.5.* For $\ell = \ell_{\exp}$,

$$\left|\frac{\ell'_{\exp}(z_1)}{\ell'_{\exp}(z_2)} - 1\right| = \left|e^{z_2 - z_1} - 1\right| \le e^{|z_2 - z_1|} - 1$$

the inequality is from $|e^x - 1| \le e^{|x|} - 1$. For $\ell = \ell_{\log}$,

$$\left|\frac{\ell'_{\log}(z_1)}{\ell'_{\log}(z_2)} - 1\right| = \left|\frac{1 + e^{z_2}}{1 + e^{z_1}} - 1\right| = \left|\frac{e^{z_2} - e^{z_1}}{1 + e^{z_1}}\right| \le \left|\frac{e^{z_2} - e^{z_1}}{e^{z_1}}\right| \le e^{|z_2 - z_1|} - 1$$

$\square$

**Lemma C.6.** For $\ell \in \{\ell_{\exp}, \ell_{\log}\}$ and any $z_1, z_2, z_3, z_4 \in \mathbb{R}$, we have

$$\left|\frac{\ell'(z_1)\ell'(z_3)}{\ell'(z_2)\ell'(z_4)} - 1\right| \le \left(e^{|z_1 - z_2|} - 1\right) + \left(e^{|z_3 - z_4|} - 1\right) + \left(e^{|z_1 + z_3 - z_2 - z_4|} - 1\right) \qquad \text{(C.3)}$$

*Proof of Lemma C.6.* For $\ell = \ell_{\exp}$,

$$\left|\frac{\ell'_{\exp}(z_1)\ell'_{\exp}(z_3)}{\ell'_{\exp}(z_2)\ell'_{\exp}(z_4)} - 1\right| = \left|e^{z_2 + z_4 - z_1 - z_3} - 1\right| \le \left(e^{|z_1 + z_3 - z_2 - z_4|} - 1\right)$$

the inequality is from $|e^x - 1| \le e^{|x|} - 1$. For $\ell = \ell_{\log}$,

$$\left|\frac{\ell'_{\log}(z_1)\ell'_{\log}(z_3)}{\ell'_{\log}(z_2)\ell'_{\log}(z_4)} - 1\right| = \left|\frac{(1 + e^{z_2})(1 + e^{z_4})}{(1 + e^{z_1})(1 + e^{z_3})} - 1\right| = \left|\frac{e^{z_2} + e^{z_4} + e^{z_2 + z_4} - e^{z_1} - e^{z_3} - e^{z_1 + z_3}}{1 + e^{z_1} + e^{z_3} + e^{z_1 + z_3}}\right|$$

$$\le \left|\frac{e^{z_2} - e^{z_1}}{e^{z_1}}\right| + \left|\frac{e^{z_4} - e^{z_3}}{e^{z_3}}\right| + \left|\frac{e^{z_2 + z_4} - e^{z_1 + z_3}}{e^{z_1 + z_3}}\right|$$

$$\le \left(e^{|z_1 - z_2|} - 1\right) + \left(e^{|z_3 - z_4|} - 1\right) + \left(e^{|z_1 + z_3 - z_2 - z_4|} - 1\right)$$

$\square$

**Lemma C.7.** For $\ell = \ell_{\log}$, and any $\mathbf{w} \in \mathbb{R}^d$, we have

$$\frac{\mathcal{G}(\mathbf{w}_t)}{\mathcal{R}(\mathbf{w})} \ge 1 - \frac{n\mathcal{R}(\mathbf{w})}{2}$$

*Proof of Lemma C.7.* Let $r_i = \ell_{\log}(\langle \mathbf{w}, \mathbf{z}_i \rangle) = \log(1 + e^{-\langle \mathbf{w}, \mathbf{z}_i \rangle})$ and $f(z) = 1 - e^{-z}$, then $|\ell'_{\log}(\langle \mathbf{w}, \mathbf{z}_i \rangle)| = \frac{e^{-\langle \mathbf{w}, \mathbf{z}_i \rangle}}{1 + e^{-\langle \mathbf{w}, \mathbf{z}_i \rangle}} = \frac{e^{r_i} - 1}{e^{r_i}} = f(z_i)$. Therefore for any given $\mathcal{R}(\mathbf{w})$, finding $\min \mathcal{G}(\mathbf{w}_t)$ equals to the following optimization problem,

$$\min \frac{1}{n}\sum_{i=1}^n f(r_i) \quad \text{s.t.} \quad \sum_{i=1}^n r_i = n\mathcal{R}(\mathbf{w}), \ r_i \ge 0 \text{ for all } i \in [n]$$

Since $f(z)$ is an increasing function and the increasing rate would be slow as $z$ increase since $f''(z) < 0$, we can easily derive that the aforementioned optimization problem will take the minimum at $r_i = n\mathcal{R}(\mathbf{w})$ for some $i \in [n]$ and $r_j = 0$ for all $j \ne i$. Therefore, we can derive that,

$$\frac{\mathcal{G}(\mathbf{w}_t)}{\mathcal{R}(\mathbf{w})} \ge \frac{1 - e^{-n\mathcal{R}(\mathbf{w})}}{n\mathcal{R}(\mathbf{w})} \ge 1 - \frac{n\mathcal{R}(\mathbf{w})}{2}$$

by Taylor's expansion. $\square$

### C.3 Auxiliary Results

The following result is the classic Stolz–Cesàro theorem.

**Theorem C.8** (Stolz–Cesàro theorem). Let $\{a_n\}_{n \ge 1}$, $\{b_n\}_{n \ge 1}$ be two sequences of real numbers. Assume that $\{b_n\}_{n \ge 1}$ is a strictly monotone and divergent sequence (i.e. strictly increasing and approaching $+\infty$, or strictly decreasing and approaching $-\infty$) and the following limit exists:

$$\lim_{n \to \infty} \frac{a_{n+1} - a_n}{b_{n+1} - b_n} = l.$$

Then it holds that

$$\lim_{n \to \infty} \frac{a_n}{b_n} = l.$$

