# OpenReview forum: "The Implicit Bias of Adam on Separable Data"
_NeurIPS.cc/2024/Conference — NeurIPS 2024 poster_

### Official Review · Reviewer_Gj1V · 2024-07-10

**Soundness:** 3
**Presentation:** 3
**Contribution:** 3
**Rating:** 6
**Confidence:** 4

**Summary:**

The main focus of this paper is on the implicit bias of Adam for a single layer linear model which performs binary classification on separable data. In particular, assuming a zero stability constant $\epsilon$, this paper reveals that Adam finds the solution that achieves maximum-$\ell_\infty$-margin and characterizes the convergence rate for different classes of learning rate. This implicit bias is different from the $\ell_2$-norm minimization solution obtained by previous work which does not assume $\epsilon = 0$.

**Strengths:**

- This paper is clearly written and well-organized. It is easy and clear to follow the argument and motivation of this paper, e.g., the proof sketch makes it easy to follow the way how the theoretical conclusion is developed. In addition, to me, the introduction of the related works are comprehensive and clear. It also clearly summarizes the difference between this paper and related works.
- The settings and results of this paper are new compared to previous works, i.e., previous works showed an $\ell_2$-norm solution implicit bias of Adam on separable data while this paper reveals an $\ell_{\infty}$-norm implicit bias when the stability constant $\epsilon$ is zero.

**Weaknesses:**

Despite the novelty of the theoretical claims, I still have several concerns, which I will discuss in the following.

1. Removing the stability constant $\epsilon$ makes the approach of this paper fails to characterize the influence of it, which, though being small, still has non-negligible effect, e.g., [1] observed that Adam with an $\epsilon$ that is too small does not even converge in certain circumstances. Treating $\epsilon$ as 0 seems a bit rough to me.

    In addition, [2] showed that Adam minimizes the interpolation norm of gradients that depends on magnitudes of various hyper parameters including the stability constant $\epsilon$ (although [2] did not specify the types of loss functions and model architectures). [1] claimed that Adam with nonzero $\epsilon$ converges to $\ell_2$-norm solution, which is also verified by extensive experiments. As a comparison, this paper showed that both Adam with $\epsilon=0$ and with a non-negligible $\epsilon$ do not converge to the aforementioned solutions (line 210). In this sense, it seems that the conclusion reached by this paper contradicts with those derived by [1, 2]. Therefore, in my view, it would be better to start with a non-zero $\epsilon$ and let the case with $\epsilon=0$ be a special case to better capture the effect of the $\epsilon$ on the implicit bias.

2. This paper only considers a simple setting: the model is only a one-layer linear model and there is no stochastic sampling noise which is typically necessary in practice. As a comparison, authors of [1] have already studied Adam on separable data for homogeneous models, which can cover the single layer model of the current work as a special case. Thus excluding the stochastic sampling noise in the current work is kind of unsatisfying to me since the model is already a simple one. In addition, I think that the authors of the current work should at least repeat the experiments conducted in [1] (such as those for homogeneous neural networks) to further support their theoretical claims, especially considering that the authors claimed in line 210 that their results are more accurate than those of [1].

**Reference**

[1] Wang et al. The implicit bias for adaptive optimization algorithms on homogeneous neural networks.

[2] Cattaneo et al. On the Implicit Bias of Adam.

**Questions:**

1. Could the authors explain the contradiction and connection with previous works? Is it possible to start with a non-zero $\epsilon$ and let $\epsilon=0$ be a special case?

2. How will adding stochastic sampling noise affect the implicit bias?

**Limitations:**

I do not find a separate limitation section in the main part. In my view, removing the stability constant is a bit rough. This makes the approach presented in this paper fail to capture how the implicit bias of Adam changes for different values of stability constant.

The societal impact is not applicable to this work as it focuses on theoretical parts of Adam.

---

> ### Author Rebuttal · Authors · 2024-08-06
>
> We appreciate your support, and address your questions as follows.
>
> >**Q1**:
> Adam without $\epsilon$ sometimes does not converge. Contradiction with [1, 2]? Study a non-zero $\epsilon$ and let $\epsilon=0$ be a special case?
>
> **A1**:
> Our goal is to study the implicit bias of Adam when the stability constant $\epsilon$ is negligible. The differences between our result and [1,2,3] that consider a non-negligible $\epsilon$ do not lead to any contradiction. Instead, these differences demonstrate that our work indeed provides novel and complementary insights into Adam. We address your detailed comments as follows:
>
> - We emphasize that Adam without $\epsilon$ provably converges under our setting. Proving convergence even without $\epsilon$ is a contribution of our work.
>
> - Our result does not contradict with [2]. By treating the correction term in the continuous-time approximation of Adam as a penalty, [2] gives an informal summary of the implicit bias of Adam without specifying the loss or model. In comparison, our work focuses on a specific setting, and provides a formal and rigorous analysis on the implicit bias of Adam with concrete convergence rates.
>
> - Our theory does not contradict with the theories in [1,3]. [1,3] prove that Adam with a non-negligible $\epsilon$ has the implicit bias towards the maximum $\ell_2$-margin. The intuition is that, after sufficiently many iterations, $\nabla \mathcal{R}(w_t)$ and $v_t$ will be very close to zero (much smaller than $\epsilon$), and $\frac{m_t}{\sqrt{v_t} + \epsilon} \approx \frac{1}{\epsilon}m_t$, indicating that Adam with a non-negligible $\epsilon$ will eventually be similar to GD with momentum. In comparison, we focus on the setting where $\epsilon$ is negligible, and demonstrate that in this case, Adam will have a unique implicit bias towards the maximum $\ell_\infty$-margin. Therefore, our theoretical results do not contradict those in [1,3]. Instead, our paper and [1,3] can together provide a more comprehensive understanding of the implicit bias of Adam.
>
> - Our experiments do not contradict with the experiments in [1,3]. In our simulations, we run different algorithms for $10^6$ iterations, and Adam can typically reach training losses below $10^{-8}$, demonstrating that it is reasonable to stop the training there. Our experiments show that within $10^6$ iterations, Adam with $\epsilon = 10^{-8}$ is indeed approaching the maximum $\ell_\infty$-margin. In comparison, in Figure 1 in [3], Adam and other algorithms are run for over $10^{14}$ iterations, where Adam with $\epsilon$ can eventually approach the maximum $\ell_2$-margin solution. Therefore, our paper focuses more on the practical stopping time, while [3] focuses more on the behavior of Adam after an extensively long training. Hence, our experiment results are complementary to [3], and there is no contradiction.
>
> - “Start with a non-zero $\epsilon$ and let $\epsilon=0$ be a special case” could be an interesting future direction, and we will discuss it in the revision. However, given that existing works [1,2,3] have studied the case with a non-zero $\epsilon$, we believe that our current setting with $\epsilon = 0$ helps us demonstrate our claim in the cleanest setting.
>
> >**Q2**:
> Linear model + full-batch Adam is simple compared with [1]. Consider stochastic sampling noise or homogeneous models?
>
> **A2**:
> Thanks for your comment. Again, our work aims to study a relatively understudied scenario where $\epsilon$ in Adam is negligible. This setting is complementary to the studies in [1,3], and therefore it is reasonable for us to start from the most classic setting to study implicit bias, which is solving linear logistic regression with full-batch algorithms [3,6].
>
> To the best of our knowledge, existing works on the implicit bias of Adam or AdamW such as [1,3,6] mainly focus on the full batch setting. Extensions to stochastic Adam can possibly be done following the analysis in [5], where the implicit bias of full batch GD is extended to SGD. However, such an extension may be more challenging for Adam due to its complex form. This can be a future direction.
>
> Extensions to homogeneous networks are also an important future direction. However, since our goal is to study the implicit bias of Adam with a negligible $\epsilon$, establishing concrete results in linear logistic regression is a reasonable starting point, and this clean and classic setting serves our purpose well in demonstrating the difference in the implicit bias of Adam when $\epsilon$ is negligible.
>
> >**Q3**:
> Line 210 claims the results are more accurate than [1], so add experiments on homogeneous networks?
>
> **A3**:
> We would like to clarify that around line 210, we do not compare our result with [1]. Instead, we are comparing with [3], which studies linear classification. We will revise our comments and highlight more on our purpose, which is to study a setting that is complementary to [1,3].
>
> We have concerns that experiments of homogeneous models are out of the scope of our paper as we do not claim anything about homogeneous models. However, we have added some preliminary experiment results in the pdf rebuttal page.
>
> [1] Wang, B., Meng, Q., Chen, W. and Liu, T.-Y. (2021). The implicit bias for adaptive optimization algorithms on homogeneous neural networks. ICML.
>
> [2] Cattaneo, M.D., Klusowski, J.M. and Shigida, B. (2024). On the Implicit Bias of Adam. ICML.
>
> [3] Wang, B., Meng, Q., Zhang, H., Sun, R., Chen, W., Ma, Z.-M. and Liu, T.-Y. (2022).
> Does momentum change the implicit regularization on separable data? NeurIPS.
>
> [4] Xie, S. and Li, Z. (2024). Implicit Bias of AdamW: $\ell_\infty $-Norm Constrained Optimization. ICML.
>
> [5] Nacson, M. S., Srebro, N. and Soudry, D. (2019). Stochastic gradient descent on separable data: Exact convergence with a fixed learning rate. AISTATS.
>
> [6] Soudry, D., Hoffer, E., Nacson, M. S., Gunasekar, S. and Srebro, N. (2018). The implicit
> bias of gradient descent on separable data. JMLR.

---

> ### Comment · Reviewer_Gj1V · 2024-08-09
> **Reply to rebuttals**
>
> I thank the authors for the detailed response.
>
> I would like to clarify that, as I pointed out in my review, I understand that the current work considers Adam without the stability constant. My point of the contradiction between the current paper and previous works lies in that how letting $\epsilon \to 0$ changes the implicit bias from the $\ell_2$ solution to the drastically different $\ell_\infty$-margin solution, e.g., is the transition smooth or abrupt? There exists a gap.
>
> The rest of the rebuttals addressed my other concerns.

---

> > ### Author Response · Authors · 2024-08-09
> >
> > Thanks for your prompt reply and for clarifying your question. We confirm that when $\epsilon\to 0^+$, the transition of the implicit bias is indeed abrupt. Consider Adam with stability constant $\epsilon$, and denote by $w_{t, \epsilon}$ its iterate at the $t$-th iteration. Then, the implicit bias of Adam with a fixed value of $\epsilon$ can be characterized by the limits:
> >
> > $\lim\_{t\to +\infty} \max\_{i\in[n]} \frac{\langle y_i\cdot x_i, w\_{t, \epsilon}\rangle}{\\|w\_{t, \epsilon}\\|\_2}$ and $\lim_{t\to +\infty} \max\_{i\in[n]} \frac{\langle y_i\cdot x_i, w\_{t, \epsilon}\rangle}{\\|w\_{t, \epsilon}\\|\_\infty}$.
> >
> > Mathematically, the abrupt transition of implicit bias when $\epsilon \to 0^+$ is then due to the fact that the limit $t\to+\infty$ and the limit $\epsilon\to0^+$ are not interchangeable:
> >
> > $\lim\_{\epsilon\to 0^+}\lim\_{t\to +\infty} \max\_{i\in[n]} \frac{\langle y_i\cdot x_i, w\_{t, \epsilon}\rangle}{\\|w\_{t, \epsilon}\\|_2} \neq \lim\_{t\to +\infty}\lim\_{\epsilon\to 0^+} \max\_{i\in[n]} \frac{\langle y_i\cdot x_i, w\_{t, \epsilon}\rangle}{\\|w\_{t, \epsilon}\\|_2}$, and
> >
> > $\lim_{\epsilon\to 0^+}\lim_{t\to +\infty} \max_{i\in[n]} \frac{\langle y_i\cdot x_i, w_{t, \epsilon}\rangle}{\\|w_{t, \epsilon}\\|\_\infty} \neq \lim_{t\to +\infty}\lim_{\epsilon\to 0^+} \max_{i\in[n]} \frac{\langle y_i\cdot x_i, w_{t, \epsilon}\rangle}{\\|w_{t, \epsilon}\\|_\infty}$.
> >
> > Therefore, there is no contradiction. We hope the discussion above can address your question.

---

> > > ### Comment · Reviewer_Gj1V · 2024-08-10
> > > **Thanks for the response**
> > >
> > > I thank the authors for the response.
> > >
> > > If the transition is abrupt as explained by the authors, then there is no contradiction, which addressed my concern on this aspect..
> > >
> > > On the other hand, from my perspective, it’s still important to clearly characterize the role of the stability constant and the abrupt transition, as well as the role stochasticity, for the reason that the setting would be more practical then.
> > >
> > > Overall, I think this paper is a good addition to the implicit bias of Adam with zero stability constant in logistic regression.

---

### Official Review · Reviewer_CKHd · 2024-07-14

**Soundness:** 3
**Presentation:** 3
**Contribution:** 3
**Rating:** 6
**Confidence:** 4

**Summary:**

This paper examines the implicit bias of the Adam optimizer in the context of linear logistic regression, demonstrating that it converges to the maximum $\ell_\infty$-margin solution under certain mild conditions. The authors note that omitting the stability constant in Adam updates results in a different implicit bias than gradient descent, with or without momentum, which converges to the maximum $\ell_2$-margin solution. They also explore various decreasing learning rates, showing that Adam's margin converges at a polynomial rate, which is faster than that of gradient descent. Additionally, they provide numerical experiments that support their findings.

**Strengths:**

- Understanding why Adam performs better than GD in several settings is an important problem and this work takes an important step towards this by showing that Adam has a different implicit bias than GD in the linear logistic regression setting.

- Overall, the paper is well-written and easy to follow. The proof sketch in Section 6 is explained well.

**Weaknesses:**

- The paper does not present results for a fixed learning rate and only considers a set of decreasing learning rates.

    - The discussion in lines 50-52 and after Corollary 4.7, comparing the rates of Adam and GD, should also comment on the convergence rates for GD with adaptive learning rates (e.g., normalized GD) which have been shown to converge faster (see [1] and related work) than GD.

    - (Minor) In Assumption 4.3, ‘non-increasing’ should be ‘decreasing’ or ‘diminishing’.

- The results in prior work on implicit bias of GD are global (hold for any initialization), whereas the results in this paper require an assumption on the initialization (Ass. 4.2). Based on the discussion following this assumption, it might be better to state an assumption on the data and then show that the condition on the initialization holds as a Lemma.

- The paper does not comment on how optimal the obtained rates in Corollary 4.7 are.

**References:**

[1] Wang et al., Achieving Margin Maximization Exponentially Fast via Progressive Norm Rescaling, 2023.

**Questions:**

Can the authors comment more on why considering the stability constant $\epsilon=0$ makes the setting more challenging? I understand the motivation in lines 105-107, but it is unclear what the challenge is since the accumulated second-order moments would be non-zero.

**Limitations:**

There is no potential negative impact of this work.

---

> ### Author Rebuttal · Authors · 2024-08-06
>
> We appreciate your positive comments. Your comments and questions are addressed as follows.
>
> >**Q1**:
> The paper does not present results for a fixed learning rate.
>
> **A1**:
> When considering fixed learning rate $\eta_t=\eta$ for some small $\eta$, our analysis can imply that $\lim_{t\to \infty}\big|\min\frac{\langle w_{t}, y_i\cdot x_i\rangle}{\lVert w_t \lVert_{\infty}}-\frac{\langle w^*, y_i\cdot x_i\rangle}{\lVert w^*\lVert_{\infty}} \big|\leq O(\sqrt{\eta})$. We will add a comment in the revision.
>
>
> >**Q2**:
> The discussion after Corollary 4.7, comparing Adam and GD, should also compare Adam and GD with adaptive learning rates (see [1] and related work).
>
> **A2**:
> Thanks for your suggestion and for pointing out the related work [1]. We will cite it and add more comprehensive comparisons and discussions in the revision.
>
> >**Q3**:
>  (Minor) In Assumption 4.3, ‘non-increasing’ should be ‘decreasing’ or ‘diminishing’.
>
> **A3**:
> Thanks for your suggestion. We will change "non-increasing" to "decreasing" in the revision. We will also add comments to clarify that we do not require the learning rates to be "strictly decreasing”.
>
> >**Q4**:
> Based on the discussion following Assumption 4.2, it might be better to state an assumption on the data and then show that the condition on the initialization holds as a Lemma.
>
> **A4**:
> We propose such an assumption because it is the most general version of assumption and it can cover various different settings. For example, we expect that the current version of Assumption 4.2 covers the following two cases:
>
> - Case 1 (fixed $w_0$, random data): consider an arbitrary fixed vector $w_0\in \mathbb{R}^d$. Then as long as the training data inputs $\mathbf{x}_1,\ldots, \mathbf{x}_n$ is sampled from any continuous and non-degenerate distribution, $\nabla \mathcal{R}(w_0)[k] \neq 0$ holds with probability $1$.
>
> - Case 2 (fixed data, random $w_0$): consider any fixed training data inputs $\mathbf{x}_1,\ldots, \mathbf{x}_n$ satisfying that the matrix $ [ \mathbf{x}_1,\ldots, \mathbf{x}_n ] $ has no all-zero row. Then as long as $w_0$ is initialized following a continuous and non-degenerate distribution,  $\nabla \mathcal{R}(w_0)[k] \neq 0$ with probability $1$.
>
> Therefore, we feel that the current version of Assumption 4.2 may be more general.
>
> Besides, we would also like to clarify that assumptions on properties of initialization have been considered in various previous works studying implicit bias [2, 3, 4, 5]. In particular, [2] makes an assumption that is essentially the same as Assumption 4.2.
>
>
> >**Q5**:
> The paper does not comment on how optimal the obtained rates in Corollary 4.7 are.
>
> **A5**:
> Currently, we do not have any lower bounds on the margin convergence rates. Due to the complicated optimization dynamics, establishing such lower bounds may be very challenging, and therefore it is difficult to prove that our rate of convergence is optimal. We will add discussions in the revision, and will also mention in the revision that establishing lower bounds on the margin convergence rates is an interesting future work direction.
>
> Despite not having matching lower bounds, our result demonstrates that the margin of the linear predictor converges in polynomial time for a general class of learning rates, which already significantly distinguishes Adam from gradient descent. Such polynomial convergence rates are also supported by our experiment results.
>
>
> >**Q6**:
> Why does considering the stability constant $\epsilon=0$ make the setting more challenging? The accumulated second-order moments would be non-zero.
>
> **A6**:
> When studying Adam without $\epsilon$, it is true that under Assumption 4.2, the gradient coordinates at initialization are non-zero and therefore the accumulated second-order moments would be non-zero throughout training. However, please note that the impact of the initial gradient values decays exponentially fast during training: in a worst case where for a certain coordinate $k$, $\nabla \mathcal{R}(w_t)[k] = 0$ for all $t > 0$, we have $\mathbf{v}_t[k] = \beta_2 (1-\beta_2)^t \cdot \mathcal{R}(w_0)[k]^2$. Therefore, without a very careful analysis, the worst-case exponentially decaying $\mathbf{v}_t[k]$ may significantly affect the stability of the algorithm, and may even impose a $\exp( t )$ factor in the bounds of convergence rate, making the bounds vacuous.
>
> In fact, many existing analyses of Adam and its variants rely on a non-zero stability constant $\epsilon$, having a factor of $1/\epsilon$ in their convergence bounds, e.g., [6] (see Theorem 4.3) and  [7] (see Corollary 1 and Remark 2). In comparison, in our work, we implement a very careful analysis (also utilizing the relatively simple problem setting of linear logistic regression) to avoid having such factors in the bounds.
>
> [1] Wang, M., Min, Z. and Wu, L. (2024). Achieving Margin Maximization Exponentially Fast via Progressive Norm Rescaling. International Conference on Machine Learning.
>
> [2] Xie, S. and Li, Z. (2024). Implicit Bias of AdamW: $\ell_\infty $-Norm Constrained Optimization. International Conference on Machine Learning.
>
> [3] Lyu, K. and Li, J. (2020). Gradient Descent Maximizes the Margin of Homogeneous Neural Networks. International Conference on Learning Representations.
>
> [4] Ji, Z. and Telgarsky, M. (2020). Directional convergence and alignment in deep learning. Advances in Neural Information Processing Systems.
>
> [5] Wang, B., Meng, Q., Chen, W. and Liu, T.Y. (2021). The Implicit Bias for Adaptive Optimization Algorithms on Homogeneous Neural Networks. International Conference on Machine Learning.
>
> [6] Zhou, D., Chen, J., Cao, Y., Yang, Z. and Gu, Q. (2024). On the Convergence of Adaptive Gradient Methods for Nonconvex Optimization. Transactions on Machine Learning Research.
>
> [7] Huang, F., Li, J. and Huang, H. (2021). SUPER-ADAM: Faster and Universal Framework of Adaptive Gradients. Advances in Neural Information Processing Systems.

---

> > ### Comment · Reviewer_CKHd · 2024-08-11
> >
> > I thank the authors for the detailed rebuttal. The lack of a discussion on the optimality of the rates presents a weakness, so I am not raising my score, but I still believe the paper makes a good contribution, so I will maintain my score.

---

### Official Review · Reviewer_dtxf · 2024-07-17

**Soundness:** 3
**Presentation:** 4
**Contribution:** 4
**Rating:** 7
**Confidence:** 3

**Summary:**

In this work, the author studies the implicit bias of Adam optimizer for a single layer neural network on separable data. The author's work suggests that, compared to the implicit bias of gradient descent which is the max $ \ell_2 $ margin solution, Adam solution converges to the maximum $ \ell_\infty $ margin solution. For this work, authors take both exponential and logistic loss and find that the convergence speed is on a polynomial order.

In order to confirm the results, the authors perform experiments on synthetic datasets for binary classification tasks and confirm Adam’s convergence to the $ \ell_\infty $ margin comparatively.

**Strengths:**

The work is novel (to the best of my knowledge) and interesting as the study of implicit bias of Adam could have further implications in characterizing the difference in optimization behavior of Adam vs SGD in practical scenarios. The assumptions of the work have been clearly presented and seem reasonable. With regard to the $ \epsilon $, while theoretical results are not provided, the authors include convincing experimental illustrations to convince me of the assumption. I also appreciate the well written proof sketch which helps convey the ideas

**Weaknesses:**

At the moment, I have some concerns with the paper which are more fit to be discussed as questions.

**Questions:**

1) Can the authors expand on how they arrive at the right side of inequality after line 292 using 6.1 ? Perhaps take me through the inequality step by step ?
2) Can the author provide some comments regarding the independence of convergence in the case of $ a = \frac{2}{3} $ from $ \rho $ ? Is there some intuition with regards to the boundaries and case on $ a $ ?

---

> ### Author Rebuttal · Authors · 2024-08-06
>
> Thank you for your supportive comments! We address them in detail as follows:
>
> >**Q1**:
> Can the authors expand on how they arrive at the right side of inequality after line 292 using 6.1 ? Perhaps take me through the inequality step by step ?
>
> **A1**:
> Thanks for your question. We would like to explain that there is a typo on line 292: "for all $t > t_2$" should be "for all $t \geq t_2$".
>
> We now present the detailed derivation as follows. Our target is to derive equation after line 292, which is
>
> $\mathcal{R}(w_{t+1}) \leq \Big(1-\frac{\gamma\eta_t}{2}\Big)\cdot \mathcal{R}(w_{t})\leq \mathcal{R}(w_{t})\cdot e^{-\frac{\gamma\eta_t}{2}}\leq \mathcal{R}(w_{t_2})\cdot e^{-\frac{\gamma\sum_{\tau=t_2}^{t}\eta_\tau}{2}}$.      $\qquad$ (target)
>
>
> To derive these results, we first recall that in equation (6.1), we have
>
> $\mathcal{R}(w_{t+1})\leq \mathcal{R}(w_{t}) - \frac{\gamma\eta_t}{2}\mathcal{G}(w_t)$.
>
> Note that in the proof sketch, for simplicity we focus on the case of exponential loss, and for the exponential loss, it holds that $\ell(x) = -\ell'(x) =e^{-x}$. Therefore, by the definition of $\mathcal{R}(w_{t})$ and $\mathcal{G}(w_{t})$, we have $\mathcal{R}(w_{t}) = \mathcal{G}(w_{t})$. Replacing $\mathcal{G}(w_{t})$ by $\mathcal{R}(w_{t})$ in equation (6.1) then gives
>
> $\mathcal{R}(w_{t+1})\leq \Big(1-\frac{\gamma\eta_t}{2}\Big)\cdot \mathcal{R}(w_{t})$.
>
> This proves the first inequality in (target). To prove the second inequality in (target), we utilize the fact that $1-x\leq e^{-x}$ holds for all $x$. Based on this, we have
>
> $1-\frac{\gamma\eta_t}{2} \leq e^{-\frac{\gamma\eta_t}{2}}$,
>
> which further implies
>
> $\Big(1-\frac{\gamma\eta_t}{2}\Big)\cdot \mathcal{R}(w_{t})\leq \mathcal{R}(w_{t})\cdot e^{-\frac{\gamma\eta_t}{2}}$.
>
> This proves the second inequality in (target). So far, we have proved that
>
> $\mathcal{R}(w_{t+1})\leq \mathcal{R}(w_{t})\cdot e^{-\frac{\gamma\eta_t}{2}}$
>
> for all $t \geq t_2$. Applying this result recursively gives
>
> $\mathcal{R}(w_{t+1})\leq \mathcal{R}(w_{t})\cdot e^{-\frac{\gamma\eta_t}{2}}  \leq \mathcal{R}(w_{t-1})\cdot e^{-\frac{\gamma\eta_{t-1}+\gamma\eta_t}{2}}\leq \cdots \leq \mathcal{R}(w_{t_2})\cdot e^{-\frac{\gamma\sum_{\tau=t_2}^{t}\eta_\tau}{2}}$.
>
> This proves the last inequality in (target), and our derivation is finished. We will clarify the derivation of this result in the revision.
>
>
> >**Q2**:
> Can the author provide some comments regarding the independence of convergence in the case of $a=2/3$ from $\rho$?
>
> **A2**:
> We believe that the margin convergence rate in the case of $a=2/3$ does depend on $\rho$, as the corresponding bound in Corollary 4.7 is
> $O(\frac{d\cdot\log t + \log n + \log \mathcal{R}(w_0) + [\log(1/\rho)]^{1/3}}{t^{1/3}})$ for exponential loss, and $O(\frac{d\cdot\log t +nd+ n \mathcal{R}(w_0) + [\log(1/\rho)]^{1/3}}{t^{1/3}})$ for logistic loss. Since we are considering the margin convergence rate as $t$ goes to infinity, these convergence rates can be simplified into $O(\frac{d\cdot\log t}{t^{1/3}})$ for both exponential and logistic losses, because the term  $ d\cdot \log t$ can eventually dominate the other terms in the numerators. However, since $d\cdot \log t$ increases very slowly in $t$ and will require exponentially many iterations to dominate the other terms, in our result we still keep the other terms to make the bounds more concrete.
>
> We guess that you are asking about why $\rho$ does not appear in the margin convergence rates for the case $a<2/3$. In fact, this is because we have applied the simplifications discussed above. Let us take exponential loss as an example. When $a<2/3$, for exponential loss, we can prove (see the bound below line 603 in Appendix B.2.) a margin convergence rate of the order
>
> $O(\frac{d t^{1-3a/2} + d^{\frac{2(1-a)}{a}} + \log n + \log \mathcal{R}(w_0) +  [\log(1/\rho)]^{1-a}}{t^{1-a}})$.
>
> It is clear that the first term in the numerator $d t^{1-3a/2}$ is the only term that increases in $t$. Moreover, as $t$ increases, the term $d t^{1-3a/2}$  will dominate the other terms in the numerator in polynomial time, which leads to the following simplification:
>
> $O(\frac{d t^{1-3a/2} + d^{\frac{2(1-a)}{a}} + \log n + \log \mathcal{R}(w_0) +  [\log(1/\rho)]^{1-a}}{t^{1-a}}) = O(\frac{d t^{1-3a/2}}{t^{1-a}}) = O(\frac{d}{t^{a/2}})$,
>
> This gives the final bound in Corollary 4.7 for $a<2/3$, which does not depend on $\rho$.
>
> >**Q3**: Is there some intuition with regards to the boundaries and case on $a$?
>
> **A3**:
> Again, we explain it for the case of exponential loss. We note that Corollary 4.7 is derived based on Theorem 4.5, in which the upper bound for margin convergence is
>
> $O(\frac{\sum_{\tau=0}^{t_0-1}\eta_\tau + d\sum_{\tau=t_0}^{t-1}\eta_\tau^{\frac{3}{2}}}{\sum_{\tau=0}^{t-1}\eta_\tau})$.
>
> Therefore, to give more concrete convergence rates when $\eta_t$ is specifically set as $(t+2)^{-a}$, we need to calculate $\sum_{\tau=0}^t \eta_\tau^{3/2}$ and $\sum_{\tau=0}^t \eta_\tau$ for different values of $a$. By properties of the series $\{ (t+2)^{-a}  \}$, this calculation can be separated into 4 cases as follows:
>
> 1. When $0<a<\frac{2}{3}$, $\sum_{\tau=0}^t \eta_\tau^{3/2}=\sum_{\tau=0}^t (\tau+2)^{-3a/2} = \Theta(t^{1-3a/2})$ and $\sum_{\tau=0}^t \eta_\tau=\sum_{\tau=0}^t \tau^{-a} = \Theta(t^{1-a})$.
>
> 2. When $a=\frac{2}{3}$, $\sum_{\tau=0}^t \eta_\tau^{3/2}=\sum_{\tau=0}^t \tau^{-3a/2} = \Theta(\log t)$ and $\sum_{\tau=0}^t \eta_\tau=\sum_{\tau=0}^t \tau^{-a} = \Theta(t^{1-a})$.
>
> 3. When $\frac{2}{3}<a<1$, $\sum_{\tau=0}^t \eta_\tau^{3/2}=\sum_{\tau=0}^t \tau^{-3a/2} = \Theta(1)$ and $\sum_{\tau=0}^t \eta_\tau=\sum_{\tau=0}^t \tau^{-a} = \Theta(t^{1-a})$.
>
> 4. When $a=1$, $\sum_{\tau=0}^t \eta_\tau^{3/2}=\sum_{\tau=0}^t \tau^{-3a/2} = \Theta(1)$ and $\sum_{\tau=0}^t \eta_\tau=\sum_{\tau=0}^t \tau^{-a} = \Theta(\log t)$.
>
> The calculations above lead to the boundary cases in Corollary 4.7.

---

> > ### Comment · Reviewer_dtxf · 2024-08-13
> >
> > Thank you very much with the detailed response. It helped clarify some of my confusions. I am happy to keep my currrent evaluation.

---

### Official Review · Reviewer_qQ6s · 2024-07-21

**Soundness:** 4
**Presentation:** 3
**Contribution:** 4
**Rating:** 7
**Confidence:** 3

**Summary:**

This paper studies the implicit bias of the Adam optimizer for logistic regression on linearly separable data. The authors prove that Adam converges to the linear classifier with the maximum $\ell_\infty$-margin. This result contrasts with the classical results on (stochastic) gradient descent (with or without momentum), which converge to the maximum $\ell_2$-margin solution.

**Strengths:**

- The authors theoretically study a popular yet not well-understood optimization method, Adam, in the context of a well-studied classical problem: logistic regression on linearly separable data. This offers a solid and insightful contribution to understanding Adam. In particular, distinguishing Adam from (S)GD with/without momentum on this classical problem is a very interesting result.
- The technical contributions are also of independent interest, as they prove the results for Adam without relying on the stability constant (which is closer to practice) and use mild assumptions.
- The paper is well-written and easy to follow. The proof sketch provides a clear and comprehensive overview of the proof of the main theorem.

**Weaknesses:**

There are no major concerns about this paper. Below are minor comments and some areas for improvement:
- The paper does not provide an intuition behind why Adam achieves the maximum $\ell_\infty$-margin solution, in contrast to GD which achieves the maximum $\ell_2$-margin solution. It would be great if the authors could offer insights on how the $\ell_\infty$-margin arises instead of the $\ell_2$-margin, for example, through a warm-up analysis with SignGD ($\beta_1=\beta_2=0$) or RMSProp ($\beta_1=0$). One way to provide an intuition is as follows: Gunasekar et al. (2018) proved that steepest descent converges to the max-margin solution, implying that SignGD (steepest descent w.r.t. $\ell_\infty$-norm) converges to the maximum $\ell_\infty$-margin solution. Since SignGD is known to be a good proxy for Adam, this may offer an insight into why Adam converges to the maximum $\ell_\infty$-margin solution.
- The authors claim that the bounds in Corollary 4.7 are derived under worst-case scenarios and argue that this is why, in practice, we often observe margins converging faster than the bounds in the corollary. However, this statement lacks supporting evidence. The paper should prove that the rate of convergence is tight. Otherwise, the observed faster convergence of margins in experiments might simply indicate that the bound is not tight enough.
- Some sentences, including those in the abstract, use the term "convergence" unclearly. For example, in the abstract, "this convergence occurs within polynomial time" does not indicate the objective (the normalized $\ell_\infty$-margin in this case) of convergence. This could be confused with other notions of convergence, such as convergence in direction (i.e., $\frac{w_t}{\lVert w_t \rVert} \to \frac{w^*}{\lVert w^* \rVert}$).
- (page 6, line 183) According to the paper, the normalized $\ell_2$-margin converges at a speed of $O(\log \log t / \log t)$ when using GD. However, this should be corrected to $O(1 / \log t)$. According to Soudry et al. (2018), the normalized weight vector converges to the maximum $\ell_2$-margin vector "in direction" with a convergence rate of $O(\log \log t / \log t)$, i.e., $\lVert \frac{w_t}{\lVert w_t \rVert} - \frac{w^*}{\lVert w^* \rVert}\rVert = O(\log \log t / \log t)$. However, the normalized $\ell_2$-margin converges at the speed of $O(1/\log t)$, i.e., $|\min \frac{\langle w_t, y_t \cdot x_t \rangle}{\lVert w_t \rVert} - \frac{\langle w^*, y_t \cdot x_t \rangle}{\lVert w^* \rVert} | = O(1/\log t)$.
- (page 1, line 25) Typo: reply on -> rely on

---
[Gunasekar et al. 2018] Characterizing Implicit Bias in Terms of Optimization Geometry, ICML 2018.

[Soudry et al. 2018] The Implicit Bias of Gradient Descent on Separable Data, JMLR 2018.

**Questions:**

- Does Theorem 4.5 imply that Adam (with a learning rate $\eta_t = (t+2)^{-a}$, $a<1$) reduces loss faster than GD (Adam: $O(e^{-\gamma t^{1-a} / 4(1-a)})$ vs. GD: $O(1/t)$)? It would be great if the authors could provide a detailed comparison of the convergence rates of loss between Adam and (S)GD with/without momentum.
- Is $\beta_1 \le \beta_2$ a necessary condition? What happens if we use Adam with $\beta_1 > \beta_2$?
- Assumption 4.4 seems to be a non-standard assumption. Is this assumption a necessary condition? Can you explain why such an assumption is needed?

**Limitations:**

The paper discusses its limitations and future directions, including the extension of the results to homogeneous neural networks and the analysis of stochastic Adam instead of full-batch Adam. I think both directions are promising avenues for future research.

---

> ### Author Rebuttal · Authors · 2024-08-06
>
> Thank you for your constructive feedback! We address your comments as follows:
>
> >**Q1**:
> The paper does not provide an intuition why Adam and GD have different implicit biases. Relation to SignGD?
>
> **A1**:
> Thanks for your suggestion. Several recent works have discussed that Adam and SignGD are closely related [2], and the implicit bias of SignGD shown by [1] can indeed provide valuable insights into the implicit bias of Adam. Some of our lemmas are also motivated by this intuition (Lemma 6.1 and Lemma A.3). We will add more discussions on this intuition in the revision.  However, we would also like to clarify that, despite the clear intuition, our proof for Adam is not a simple combination of existing techniques. For example, the proofs of Lemma 6.1 and Lemma A.3 are non-trivial, and in Lemma A.3, we have implemented a particularly careful analysis which enables us to cover general learning rates and general values of $\beta_1,\beta_2$.
>
>
> >**Q2**:
> The authors claim that Corollary 4.7 is derived under the worst case and this is why in experiments we often observe margins converging faster than the bounds. This statement lacks supporting evidence. The paper should prove that the rate of convergence is tight.
>
> **A2**:
> Currently, we do not have any lower bounds on the margin convergence rates. Due to the complicated optimization dynamics, establishing such lower bounds may be very challenging, and therefore it is difficult to prove that our rate of convergence is optimal. We will make it clear in the revision. We will also mention in the revision that establishing lower bounds on the margin convergence rates is an interesting future work direction.
>
> Despite not having matching lower bounds, our result demonstrates that the margin of the linear predictor converges in polynomial time for a general class of learning rates, which already significantly distinguishes Adam from gradient descent. Such polynomial convergence rates are also supported by our experiment results.
>
> >**Q3**:
> The term "convergence" is used unclearly: objective convergence vs normalized margin convergence.
>
> **A3**:
> Thanks for pointing out the unclear expressions. We will revise them.
>
> >**Q4**: on page 6, line 183, $\ell_2$-margin convergence speed for GD should be $O(1/\log t)$.
>
> **A4**:
> Thanks for pointing it out. We will clarify it in the revision.
>
> >**Q5**:
> (page 1, line 25) Typo: reply on -> rely on
>
> **A5**:
> Thanks for pointing out this typo. We will fix it.
>
> >**Q6**:
> Does Theorem 4.5 imply that Adam reduces loss faster than GD? The authors should provide a detailed comparison.
>
> **A6**:
> Thanks for your suggestion. You are right that our result shows an $O(e^{-\frac{\gamma t^{1-a}}{4(1-a)}})$ convergence rate of the training loss for Adam in logistic regression when the data are linearly separable, and this is faster than the $O(1/t)$ convergence rate of gradient descent. In the revision, we will comment on it, and give a detailed comparison of the convergence rates of loss between Adam and (S)GD with/without momentum.
>
> >**Q7**:
> Is $\beta_1\leq\beta_2$ necessary? What happens if we use Adam with $\beta_1>\beta_2$?
>
> **A7**:
> Yes, this condition $\beta_1\leq\beta_2$ is necessary in our analysis. Such a condition ensures the stability of Adam. Under this condition, we can show that in each iteration of Adam, the update in each coordinate is bounded by a constant (Lemma 6.5).
>
> On the other hand, without this condition, there exist extreme cases where Adam is unstable. For example, consider $\beta_2=0$, $\beta_1 > 0$, and suppose that there exists a coordinate index $k$ such that at iteration $t = 1$, the corresponding coordinate of the gradient is very close to zero, satisfying $0 \leq \nabla \mathcal{R}(w_1)[k] \leq \beta_1(1-\beta_1)\rho / 10000 $. Then, by Assumption 4.2, we can see that $\mathbf{m}_1[k] \geq \beta_1(1-\beta_1)\rho$, and hence
> $ \mathbf{m}_1[k] / \sqrt{\mathbf{v}_1[k]} \geq \frac{ \beta_1(1-\beta_1)\rho }{ \beta_1(1-\beta_1)\rho / 10000 } = 10000 $. Clearly, $\mathbf{m}_1[k] / \sqrt{\mathbf{v}_1[k]}$ can be arbitrarily large when $\nabla \mathcal{R}(w_1)[k]$ tends to zero. This implies that when $\beta_2=0$, $\beta_1 > 0$, there are cases where Adam is unstable. For general $\beta_1,\beta_2$ with $\beta_1 > \beta_2$, there are also certain extreme scenarios where Adam is unstable.
>
> We would also like to point out that the condition $\beta_1\leq\beta_2$ is a common condition considered in recent works studying Adam [3, 4, 5]. Besides, this condition aligns well with practice, where popular choices are ($\beta_1=0.9, \beta_2=0.99$), or ($\beta_1=0.9, \beta_2=0.999$). We will add more explanations in the revision.
>
>
> >**Q8**:
> Assumption 4.4 seems non-standard. Is it a necessary condition? Why is such an assumption needed?
>
> **A8**:
> Assumption 4.4 is necessary for our proof. Our proof of Lemma 6.1 relies on this assumption (line 482-line 484, line 485-line 488). As we commented below Assumption 4.4 and proved in Lemma C.1, this assumption is quite mild: it holds for fixed (small enough) learning rate, or decaying learning rates $\eta_t = (t+2)^{-a}$ with $a \in (0, 1]$.
>
> [1] Gunasekar, S., Lee, J., Soudry, D. and Srebro, N. (2018). Characterizing implicit bias in terms of optimization geometry. International Conference on Machine Learning.
>
> [2] Balles, L., Pedregosa, F., and Roux, N. L. (2020). The geometry of sign gradient descent. arXiv preprint arXiv:2002.08056.
>
> [3] Xie, S. and Li, Z. (2024). Implicit Bias of AdamW: $\ell_\infty $-Norm Constrained Optimization. International Conference on Machine Learning.
>
> [4] Hong, Y. and Lin, J. (2024). On Convergence of Adam for Stochastic Optimization under Relaxed Assumptions. arXiv preprint arXiv:2402.03982.
>
> [5] Zou, D., Cao, Y., Li, Y. and Gu, Q. (2023). Understanding the generalization of adam in learning neural networks with proper regularization. International Conference on Learning Representations.

---

> > ### Comment · Reviewer_qQ6s · 2024-08-08
> >
> > Thank you for the clarifications. I have thoroughly reviewed the rebuttal and have no further questions. I appreciate the authors' efforts and am happy to maintain my score, voting for acceptance. I look forward to seeing the discussions mentioned in the rebuttal incorporated into the revised manuscript.

---

> > > ### Author Response · Authors · 2024-08-09
> > >
> > > Thank you for your support and detailed suggestions. We will make sure to include our discussions in the revised version of the paper.

---

### Author Rebuttal · Authors · 2024-08-06

Dear Reviewers,

We appreciate your supportive and constructive comments on our paper. We have addressed all your questions in detail in our individual responses to you. Here, as suggested by Reviewer Gj1V, we include a pdf page presenting some preliminary experiment results on training homogeneous neural networks. But we would also like to clarify that we do not aim to claim any conclusions about trianing neural networks -- our work is still focused on the simple and clean setting of linear classification.

We lookfoward to your further comments and suggestions, and we are happy to discuss any remaining questions in detail.

Best regards,

Authors

---

### Decision · Program_Chairs · 2024-09-25

**Decision:**

Accept (poster)

**Comment:**

The paper presents a significant contribution to the understanding of Adam’s implicit bias, particularly in a setting where the stability constant is negligible. While there are some limitations due to the simplified setting and the exclusion of the stability constant, the paper’s novel theoretical insights and thorough analysis make it a valuable addition to the literature. The reviewers unanimously recommended to accept this paper. The authors should incorporate the feedbacks of reviewers into the camera-ready version.